# What is the Solution for State-Adversarial Multi-Agent Reinforcement Learning?

**Songyang Han**  *songyang.han@uconn.edu*
*School of Computing*
*University of Connecticut*
*Sony AI*

**Sanbao Su**  *sanbao.su@uconn.edu*
*School of Computing*
*University of Connecticut*

**Sihong He**  *sihong.he@uconn.edu*
*School of Computing*
*University of Connecticut*

**Shuo Han**  *hanshuo@uic.edu*
*Department of Electrical and Computer Engineering*
*University of Illinois Chicago*

**Haizhao Yang**  *hzyang@umd.edu*
*Department of Mathematics and Department of Computer Science*
*University of Maryland College Park*

**Shaofeng Zou**  *szou3@buffalo.edu*
*Department of Electrical Engineering and Department of Computer Science and Engineering*
*University at Buffalo, The State University of New York*

**Fei Miao**  *fei.miao@uconn.edu*
*School of Computing*
*University of Connecticut*

**Reviewed on OpenReview:** *https://openreview.net/forum?id=HyqSwNhM3x)*

## Abstract

Various methods for Multi-Agent Reinforcement Learning (MARL) have been developed with the assumption that agents' policies are based on accurate state information. However, policies learned through Deep Reinforcement Learning (DRL) are susceptible to adversarial state perturbation attacks. In this work, we propose a State-Adversarial Markov Game (SAMG) and make the first attempt to investigate different solution concepts of MARL under state uncertainties. Our analysis shows that the commonly used solution concepts of optimal agent policy and robust Nash equilibrium do not always exist in SAMGs. To circumvent this difficulty, we consider a new solution concept called robust agent policy, where agents aim to maximize the worst-case expected state value. We prove the existence of robust agent policy for finite state and finite action SAMGs. Additionally, we propose a Robust Multi-Agent Adversarial Actor-Critic (RMA3C) algorithm to learn robust policies for MARL agents under state uncertainties. Our experiments demonstrate that our algorithm outperforms existing methods when faced with state perturbations and greatly improves the robustness of MARL policies. Our code is public on `https://songyanghan.github.io/what_is_solution/`.

# 1   Introduction

Multi-Agent Reinforcement Learning (MARL) has been successfully used to solve problems such as multi-robot coordination (Hüttenrauch & Šošić, 2017), resource management (Pretorius et al., 2020), etc. However, Deep Reinforcement Learning (DRL) policies are vulnerable to adversarial state perturbation attacks (Behzadan & Munir, 2017; Pattanaik & Tang, 2017; Huang et al., 2017; Lin et al., 2017; Xiao et al., 2019). Even small changes to the state can lead to drastically different actions (Huang et al., 2017; Lin et al., 2017). To address this, it is important to develop robust policies that can handle adversarial state perturbations. An example of this is shown in Fig. 1 where agents need to cooperate and avoid collisions while occupying landmarks. In (a) with no adversarial state, the agents are able to target different landmarks, but in (b) with adversarial state perturbations, agents head in the wrong direction.

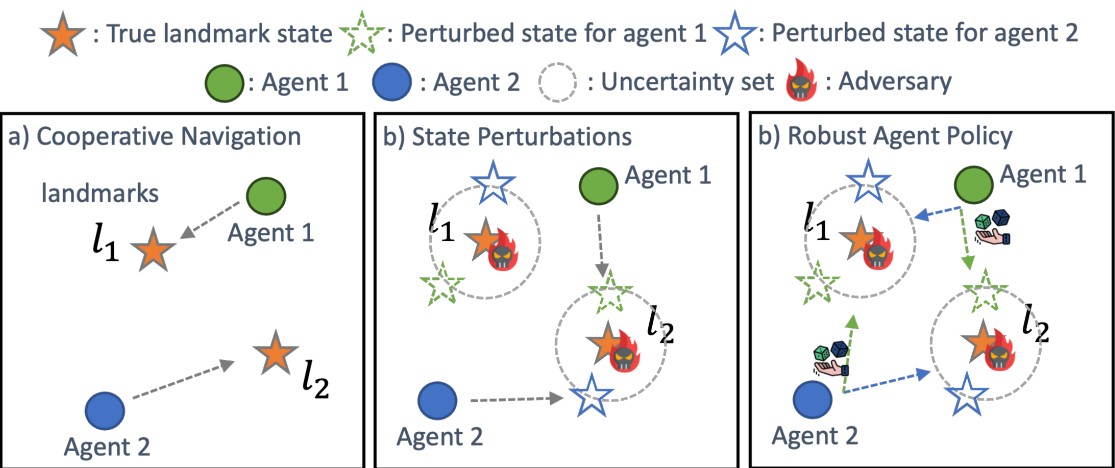

Figure 1: The agents' goal is to occupy and cover all landmarks, requiring cooperation to decide which landmark to cover. Figure a) illustrates the optimal target landmark for each agent without state perturbation. However, in figure b), an adversary perturbs the state observation of agents, causing agents to head in the wrong direction and leaving landmark 1 as uncovered. Our work demonstrates that traditional agent policies can be easily corrupted by adversarial state perturbations. To counter this, we propose a robust agent policy that maximizes average performance under worst-case state perturbations.

The adversarial state perturbation problem cannot be fully understood using existing research on the Partially Observable Markov Decision Process (POMDP) or Decentralized Partially Observable Markov Decision Process (Dec-POMDP) (Oliehoek et al., 2016; Lerer et al., 2020), as the conditional observation probability cannot capture the worst-case uncertainty under adversarial attacks. Adversarial perturbations have a greater impact on an agent's policy than random noise (Kos & Song, 2017; Pattanaik et al., 2018). However, due to the complexity of interactions among agents and adversaries, it remains challenging to formally analyze the existence of optimal or equilibrium solutions under adversarial state perturbations in MARL. Therefore, it is essential to study the fundamental properties of MARL under adversarial state perturbations.

To the best of our knowledge, we make the first attempt to investigate different solution concepts of robust MARL under adversarial state perturbations. We formulate a State-Adversarial Markov Game (SAMG) to study the properties and solution concepts of MARL under adversarial state perturbations. We prove that a state-robust totally optimal agent policy or robust total Nash equilibrium does not always exist in such scenarios. Instead, we consider a new solution concept, the robust agent policy, and prove its existence for finite state and action spaces. We design an algorithm, called Robust Multi-Agent Adversarial Actor-Critic (RMA3C), to train robust policies for all agents under adversarial state perturbations. The algorithm uses a Gradient Descent Ascent (GDA) optimizer to update each agent's and adversary's policy network. Results from our experiments show that the proposed RMA3C algorithm improves the robustness of the agents' policies compared to existing MARL methods.

In summary, the main contributions of this work are:

- We study the fundamental properties of MARL under adversarial state perturbations and prove that widely used solution concepts such as optimal agent policy or robust Nash equilibrium do not always exist.

- We consider a new solution concept, robust agent policy, where each agent aims to maximize the worst-case expected state value. We prove the existence of a robust agent policy for SAMGs with finite state and action spaces. We propose a Robust Multi-Agent Adversarial Actor-Critic (RMA3C) algorithm to solve the challenge of training robust policies under adversarial state perturbations based on gradient descent ascent algorithm.

- We empirically evaluate our proposed RMA3C algorithm. Our algorithm outperforms baselines with random or adversarial state perturbations and improves agent policies' robustness under state uncertainties.

## 2 Related Work

**Multi-Agent Reinforcement Learning (MARL)** The MARL has a long history in the AI field (Littman, 1994; Hu et al., 1998; Busoniu et al., 2008). Recent works have been investigated to encourage the collaboration of the agents by assigning rewards appropriately, such as a value decomposition network (Sunehag et al., 2018; Rashid et al., 2020; Su et al., 2021), subtracting a counterfactual baseline (Foerster & Farquhar, 2018), or an implicit method (Zhou et al., 2020). Multi-Agent Deep Deterministic Policy Gradient (MADDPG) proposes a centralized Q-function to alleviate the problem caused by the non-stationary environment (Lowe et al., 2017). The scalability issue of MARL can be alleviated by adding attention to the critic (Iqbal & Sha, 2019), using neighbor information (Qu et al., 2020), or using V-learning (Jin et al., 2021). The "team stochastic game" (Muniraj et al., 2018; Phan et al., 2020) splits the MARL agents into two teams to compete. However, during training, all methods assume that agents get the true state value. None of the recent MARL advances specifies how to deal with perturbed state values by malicious adversaries.

**Robust Reinforcement Learning** Most existing robust MARL works focus on uncertainties in reward, transition dynamics, and training partners' policies, while our work focuses on uncertainties in the state. Robust reinforcement learning can be traced back to Morimoto & Doya (2005) in the single-agent setting. With the advent of deep learning techniques, the robust MARL has been recently studied considering different types of uncertainties such as reward (Chen & Bowling, 2012; Zhang et al., 2020b), transition dynamics (Zhang et al., 2020b; Sinha et al., 2020; Hu et al., 2020; Yu et al., 2021; Wang et al., 2023), training partner's type (Shen & How, 2021), training partners' policies (Li et al., 2019; van der Heiden et al., 2020; Sun et al., 2021; 2022). The work in (Zhang et al., 2020b) considers the robust equilibrium of multi-agents with reward uncertainties where agents can access true state information at each stage. The work in Shen & How (2021) considers uncertain training partner's type (e.g. adversary, neutral, or teammate) to the protagonist in two-player scenarios. The M3DDPG algorithm extends the MADDPG to get a robust policy for the worst situation by assuming all the training partners are adversaries (Li et al., 2019). However, none of the above MARL works consider the state perturbations.

For adversarial state perturbations, there are some works (Mandlekar et al., 2017; Pinto et al., 2017; Pattanaik et al., 2018; Zhang et al., 2020a; 2021; Liang et al., 2022) considering a robust policy in single-agent reinforcement learning. Though the work (Lin et al., 2020a) studies state perturbation, only one single agent's state observation can be perturbed in their MARL. The work (He et al., 2023) shows Nash equilibrium exists under a specific condition (bijective mapping for adversary policies). However, in this work, we show the Nash equilibrium is not a good solution concept as it can be corrupted by state perturbation adversaries. We also propose a new robust agent policy concept for state-adversarial MARL that is proven to exist.

## 3 State-Adversarial Markov Game (SAMG)

We formulate a State-Adversarial Markov Game (SAMG) $G = (\mathcal{N}, \mathcal{S}, \mathcal{A}, r, \mathcal{P}_s, p, \gamma, \Pr(s_0))$ with $n$ agents in the agent set $\mathcal{N} = \{1, ..., n\}$. Each agent $i$ is associated with an action $a^i \in \mathcal{A}^i$. The global joint action is $a = (a^1, ..., a^n) \in \mathcal{A}$, $\mathcal{A} := \mathcal{A}^1 \times \cdots \times \mathcal{A}^n$. The global joint state is $s \in \mathcal{S}$. The probability distribution of the

initial state is $\Pr(s_0)$. All agents share a stage-wise reward function $r : \mathcal{S} \times \mathcal{A} \to \mathbb{R}$. We consider that each agent is associated with an adversary as shown in Fig. 2. Each adversary decides a perturbed state $\rho^i \in \mathcal{S}$ for the corresponding agent as the agent's perturbed knowledge or observation about the global state. We denote the joint perturbed state as $\rho := [\rho^i]_{i \in \mathcal{N}}$. We consider the admissible perturbed state as a task-specific "neighboring" state of $s$, e.g. the bounded sensor measurement errors, to model the challenges of getting accurate states for multi-agent systems like connected and autonomous vehicles and multi-robots systems (Liu et al., 2021; Kothandaraman et al., 2021). To analyze a realistic problem, the power of the state perturbation should also be limited (Everett et al., 2021; Zhang et al., 2020a). We define an admissible perturbed state set $\mathcal{P}_s$ to restrict the perturbed state only to be within a predefined subset of states such that $\rho \in \mathcal{P}_s$:

**Definition 3.1 (Admissible Perturbed State Set).** We consider the set of admissible perturbed state for agent $i$ at state $s$ as $\mathcal{P}_s^i \subseteq \mathcal{S}$. Denote the joint admissible perturbed state set at state $s$ as $\mathcal{P}_s := \mathcal{P}_s^1 \times \cdots \times \mathcal{P}_s^n$.

Note that the true state is included in the admissible perturbed state set, i.e., $s \in \mathcal{P}_s^i$ for any $i \in \mathcal{N}$. For example, consider a 2-agent 3-state system with $\mathcal{S} = \{s_1, s_2, s_3\}$. When the current true state is $s_1$ for both agents, adversary 1 perturbs agent 1's state observation within $\mathcal{P}_{s_1}^1 = \{s_1, s_2\}$; adversary 2 perturbs agent 2's state observation within $\mathcal{P}_{s_1}^2 = \{s_1, s_3\}$.

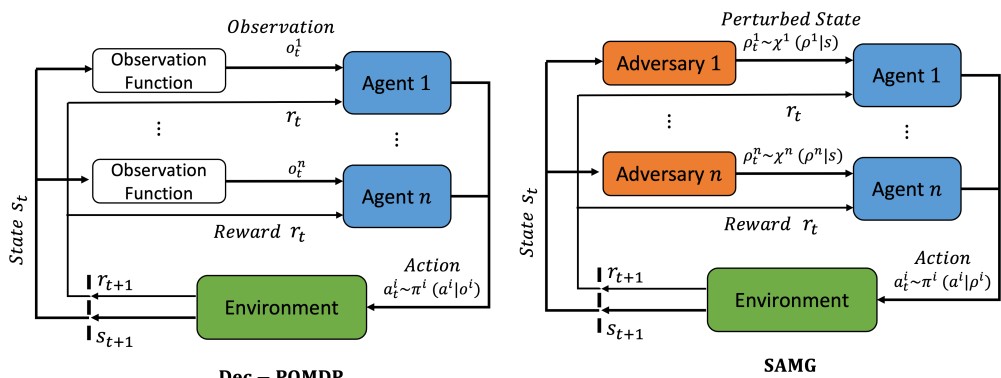

Figure 2: Comparison between Dec-POMDP and SAMG. In Dec-POMDP, the observation probability function is fixed, and it will not change according to the change of the agent policy. However, in SAMG the adversary policy is not a fixed policy, it may change according to the agents' policies and always select the worst-case state perturbation for agents. In SAMG, each agent is associated with an adversary to perturb its knowledge or observation of the true state. Agents want to find a policy $\pi$ to maximize their total expected return while adversaries want to find a policy $\chi$ to minimize agents' total expected return.

The state perturbation reflects the state uncertainty from the perspective of each agent, but it does not change the true state of multi-agent systems. The state transition function is $p : \mathcal{S} \times \mathcal{A} \to \Delta(\mathcal{S})$, where $\Delta(\mathcal{S})$ is a probability simplex denoting the set of all possible probability measures on $\mathcal{S}$. The state still transits from the true state to the next state. Each agent is associated with a policy $\pi^i : \mathcal{S} \to \Delta(\mathcal{A}^i)$ to choose an action $a^i \in \mathcal{A}^i$ given the perturbed state $\rho^i$. Note that the input of $\pi^i$ is the perturbed state $\rho^i$. The perturbed state affects each agent's action. The set $\Delta(\mathcal{A}^i)$ includes all possible probability measures on $\mathcal{A}^i$. We use $\pi = (\pi^1, \pi^2, ..., \pi^n)$ to denote the joint agent policy.

The adversary policy, i.e. the state perturbation policy, associated with agent $i$ is $\chi^i(\cdot|s) : \mathcal{S} \to \Delta(\mathcal{P}_s^i)$, where the input of $\chi^i$ is the true state $s \in \mathcal{S}$. The power of the adversary is limited by the admissible perturbed state set $\mathcal{P}_s^i$. We denote the joint adversary policy as $\chi = (\chi^1, \chi^2, ..., \chi^n)$. The agents want to find a policy $\pi$ to maximize their total expected return while adversaries want to find a policy $\chi$ to minimize the agents' total expected return. The total expected return is $\mathbb{E}[\sum_{t=0}^{\infty} \gamma^t r_{t+1}(s_t, a_t)|s_0, a_t \sim \pi(\cdot|\rho_t), \rho_t \sim \chi(\cdot|s_t)]$ where $\gamma$ is a discount factor.

Our SAMG problem cannot be solved by the existing work for single-agent RL with adversarial state perturbations (Mandlekar et al., 2017; Pattanaik et al., 2018; Zhang et al., 2020a; 2021; Liang et al., 2022). Each agent's action in SAMG is selected based on its own perturbed state observation and the state knowledge

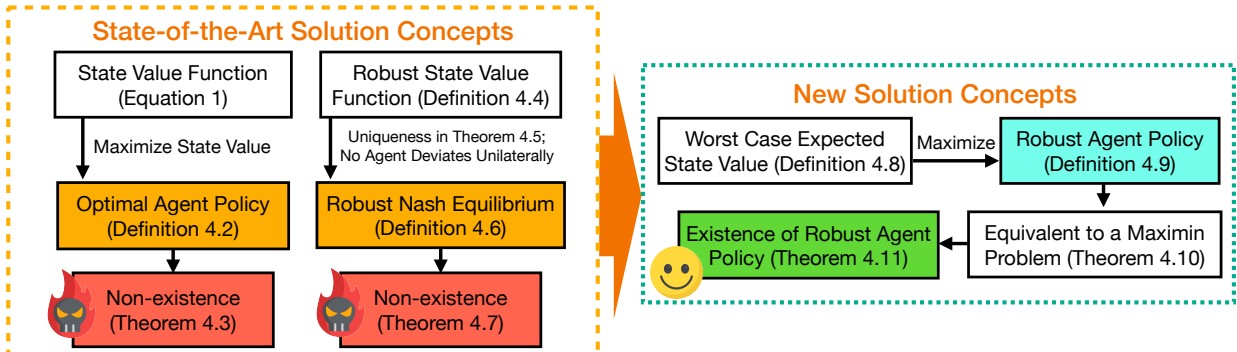

Figure 3: Solution concepts for the SAMGs. We first examine the widely used concepts (optimal agent policy and robust Nash Equilibrium) and demonstrate their non-existence under adversarial state perturbations. In response, we consider a new objective, the worst-case expected state value, and a new solution concept, the robust agent policy.

of each agent can be different after adversarial perturbations, so the SAMG problem cannot be solved by the above single-agent RL where the agent has only one state observation at each stage.

Our SAMG problem cannot be solved by the existing work in the Decentralized Partially Observable Markov Decision Process (Dec-POMDP) (Bernstein et al., 2002; Oliehoek et al., 2016) as shown in Fig. 2. In contrast, the policy in SAMG needs to be robust under a set of admissible perturbed states. The adversary aims to find the worst-case state perturbation policy $\chi$ to minimize the MARL agents' total expected return, but the Dec-POMDP cannot characterize the worst-case state perturbations. Moreover, agents usually cannot get the true state $s$ in Dec-POMDP, while in SAMG, the true state $s$ is known by the adversaries. Adversaries can take the true state information and use it to select state perturbations for the MARL agents. The following proposition 3.2 shows that under a fixed adversarial policy, the SAMG problem becomes a Dec-POMDP. However, in SAMG the adversary policy is not a fixed policy, it may change according to the agents' policies (see Theorem 4.1 for detail) and always select the worst-case state perturbation for agents. The proof of proposition 3.2 is in Appendix A. We also give a two-agent two-state SAMG that cannot be solved by Dec-POMDP in Appendix A.

**Proposition 3.2.** *When the adversary policy is a fixed policy, the SAMG problem becomes a Dec-POMDP (Oliehoek et al., 2016).*

**Proposition 3.3.** *When the adversary policy is a fixed bijective mapping from $\mathcal{S}$ to $\mathcal{S}$, the SAMG problem becomes a Markov game.*

Additionally, our SAMG problem cannot be solved by existing methods for robust Markov games considering the uncertainties from reward (Chen & Bowling, 2012; Zhang et al., 2020b), transition dynamics (Zhang et al., 2020b; Hu et al., 2020; Sinha et al., 2020; Yu et al., 2021; Wang et al., 2023), training partner's policies (Li et al., 2019; van der Heiden et al., 2020). These methods are not applicable to our problem because the agents do not have access to the true state information after adversarial perturbations.

## 4 Solution Concepts

In this section, we delve into the solution concepts of the SAMG. We formally define key concepts such as an optimal adversary policy, state-robust totally optimal agent policy, and robust total Nash equilibrium. However, we also demonstrate that under an optimal adversary policy, the existence of a state-robust totally optimal agent policy or robust total Nash equilibrium is not guaranteed as they can be easily corrupted by adversaries. Therefore, we introduce a new objective, the worst-case expected state value, and prove that there exists a robust agent policy to maximize it. A concept diagram of this section is shown in Fig. 3.

We first introduce the widely used state value function concept for our proposed SAMG as follows:

$$V_{\pi,\chi}(s) = \mathbb{E}_{a_t \sim \pi(\cdot|\rho_t), \rho_t \sim \chi(\cdot|s_t)} \left[ \sum_{t=0}^{\infty} \gamma^t r_{t+1}(s_t, a_t)|s_0 = s \right], \tag{1}$$

where $\gamma$ is the discount factor.

## 4.1 Optimal Adversary Policy

For a fixed agent policy $\pi$, define the worst-case state value function $\bar{V}_\pi$ under $\pi$ by

$$\bar{V}_\pi(s) := \min_\chi V_{\pi,\chi}(s) \tag{2}$$

for all $s \in \mathcal{S}$. An adversary policy $\chi^*$ is said to be *optimal* against an agent policy $\pi$ if

$$V_{\pi,\chi^*}(s) = \bar{V}_\pi(s) \tag{3}$$

for all $s \in \mathcal{S}$. The following proposition shows the existence of an optimal adversary for an SAMG.

**Proposition 4.1** (**Existence of Optimal Adversary Policy**.). *Given an SAMG, for any given agent policy, there exists an optimal adversary policy.*

The key process of the proof in Appendix B.4 is constructing an MDP for the adversary where the adversary gets the negative of the agent reward. Since for an MDP with finite state and finite action spaces, there always exists an optimal policy [Theorem 6.2.10 in Puterman (2014)], an optimal adversary policy of the corresponding SAMG always exists as well.

## 4.2 State-robust Totally Optimal Agent Policy

An optimal adversary policy is very powerful and it can easily corrupt the MARL agents' policies through state perturbations. We first define a state-robust totally optimal agent policy as follows:

**Definition 4.2** (**State-robust Totally Optimal Agent Policy**). An agent policy $\pi^*$ is a state-robust totally optimal agent policy if $\bar{V}_{\pi^*}(s) \geq \bar{V}_\pi(s)$ for any $\pi$ and all $s \in \mathcal{S}$.

In the following theorem, we show that a state-robust totally optimal agent policy $\pi^*$ does not always exist for SAMGs under an optimal state perturbation adversary.

**Theorem 4.3** (**Non-existence of State-robust Totally Optimal Agent Policy**). *A state-robust totally optimal agent policy does not always exist for SAMGs.*

The proof in Appendix B.5 is done by constructing a counterexample where there is no optimal policy for the agents. A state-robust totally optimal agent policy is expected to maximize the state value for all states. However, under the adversarial state perturbations, sometimes agents have to make trade-offs between different state values and no agent policy can maximize all the state values.

## 4.3 Robust Total Nash Equilibrium

Then we look at the widely-used Nash equilibrium concept in MARL for SAMGs. A Nash equilibrium is used to describe policies where no agent wants to deviate unilaterally. If an agent deviates from a Nash equilibrium, its total expected return won't increase. Denote the agent policies and adversary policies of all other agents and adversaries except agent $i$ and adversary $i$ as $\pi^{-i}$ and $\chi^{-i}$ respectively. Before giving the definition of a robust total Nash equilibrium, we first show that there exists a unique robust state value function for agent $i$ given any $\pi^{-i}$ and $\chi^{-i}$.

**Definition 4.4** (**Robust state value function**). A state value function $V^i_{*,\pi^{-i},*,\chi^{-i}} : \mathcal{S} \to \mathbb{R}$ for agent $i$ given $\pi^{-i}$ and $\chi^{-i}$ is called a robust state value function if for all $s \in \mathcal{S}$,

$$V^i_{*,\pi^{-i},*,\chi^{-i}}(s) = \max_{\pi^i} \min_{\chi^i} \sum_{\rho \in \mathcal{P}_s} \chi(\rho|s) \sum_{a \in \mathcal{A}} \pi(a|\rho) \left( r(s,a) + \gamma \sum_{s' \in \mathcal{S}} p(s'|s,a) V^i_{*,\pi^{-i},*,\chi^{-i}}(s') \right).$$

**Theorem 4.5** (**Existence of Unique Robust State Value Function**). *For an SAMG with finite state and finite action spaces, for any $i \in \mathcal{N}$, given any $\pi^{-i}$ and $\chi^{-i}$ of other agents and adversaries except agent $i$*

*and adversary $i$, there exists a unique robust state value function $V^i_{*,\pi^{-i},*,\chi^{-i}} : \mathcal{S} \to \mathbb{R}$ for agent $i$ such that for all $s \in \mathcal{S}$,*

$$V^i_{*,\pi^{-i},*,\chi^{-i}}(s) = \max_{\pi^i} \min_{\chi^i} \sum_{\rho \in \mathcal{P}_s} \chi(\rho|s) \sum_{a \in \mathcal{A}} \pi(a|\rho) \left( r(s,a) + \gamma \sum_{s' \in \mathcal{S}} p(s'|s,a) V^i_{*,\pi^{-i},*,\chi^{-i}}(s') \right).$$

The proof is based on the contraction mapping property and Banach's fixed point theorem in Appendix C.1. The theorem could also follow from viewing the SAMG as an extensive form game under imperfect recall and applying the approaches used to analyze those games Chen & Bowling (2012).

Based on the well-defined robust state value function, a robust total Nash equilibrium is defined considering each agent is associated with an adversary that tries to minimize its total expected return.

**Definition 4.6** (**Robus Total Nash Equilibrium**). For an SAMG, the policy $(\pi^*, \chi^*)$ is a robust total Nash equilibrium if for all $s \in \mathcal{S}$ and all $i \in \mathcal{N}$ and all $\pi^i$ and $\chi^i$, it holds that

$$V^i_{\pi^i,\pi^{-i*},\chi^{i*},\chi^{-i*}}(s) \leq V^i_{\pi^{i*},\pi^{-i*},\chi^{i*},\chi^{-i*}}(s) \leq V^i_{\pi^{i*},\pi^{-i*},\chi^i,\chi^{-i*}}(s), \tag{4}$$

where $\pi^{-i}$ and $\chi^{-i}$ denotes the agent policies and adversary policies of all the other agents except agent $i$, respectively.

Definition 4.6 shows that $\pi^*$ is in a robust total Nash equilibrium if each agent's policy is a robust best response to the other agents' policies under adversarial state perturbations. When agent $i$ is calculating its robust best response, it assumes a worst-case perspective of the state perturbations.

**Theorem 4.7** (**Non-existence of Robust Total Nash Equilibrium**). *For SAMGs with finite state and finite action spaces, the robust total Nash equilibrium defined in Definition 4.6 does not always exist.*

The proof in Appendix C.3 is done by constructing a counterexample. For any state $s \in \mathcal{S}$, there exists a stage-wise equilibrium among the agents and adversaries (See the proof for a stage-wise equilibrium in Theorem C.6 in Appendix C.2). However, due to the uncertainty of the true state under adversarial state perturbations, it is possible that the stage-wise equilibrium in one state conflicts with the stage-wise equilibrium in another state. As a result, the agents may be required to make trade-offs between different states, making it impossible to find an equilibrium that holds for all states.

## 4.4 Robust Agent Policy

The state-robust totally optimal agent policy and robust total Nash equilibrium concepts do not always exist in our SAMG problem according to the above non-existence analysis. To circumvent this difficulty, we consider another solution concept in which each agent adopts a policy (hereafter referred to as a *robust agent policy*) that maximizes the reward under the worst-case state perturbation. We further show that a robust policy always exists for all agents. We first introduce a new objective for SAMGs, the worst-case expected state value:

**Definition 4.8** (**Worst-case Expected State Value**). The worst-case expected state value under the optimal state perturbation adversary is: $\mathbb{E}_{s_0 \sim \Pr(s_0)} \left[ \bar{V}_\pi(s_0) \right]$, where $\Pr(s_0)$ is the probability distribution of the initial state.

To account for the fact that a policy is not able to maximize all state values in SAMG, we can use the probability of each state as a measure of its importance, and balance the values of different states. The worst-case expected state value is calculated by taking a weighted sum of all state values based on their initial state distribution. The agent policy that aims to maximize this worst-case expected state value is referred to as a robust agent policy.

**Definition 4.9** (**Robust Agent Policy**). An agent policy $\pi^*$ that maximizes the worst-case expected state value is called a robust agent policy:

$$\pi^* \in \arg\max_{\pi} \mathbb{E}_{s_0 \sim \Pr(s_0)} \left[ \bar{V}_\pi(s_0) \right]. \tag{5}$$

The following theorem shows finding a robust agent policy is equivalent to solving a maximin problem.

**Theorem 4.10.** *Finding an agent policy $\pi$ to maximize the worst-case expected state value under an optimal adversary for $\pi$ is equivalent to the maximin problem:* $\max_\pi \min_\chi \sum_{s_0} \Pr(s_0) V_{\pi,\chi}(s_0)$.

In the following theorem, we show the existence of a robust agent policy for finite state and finite action spaces.

**Theorem 4.11** (**Existence of Robust Agent Policy**)**.** *For SAMGs with finite state and finite action spaces, there exists a robust agent policy to maximize the worst-case expected state value defined in Definition 4.8.*

The proof in Appendix C.4 is based on the Weierstrass M-test (Rudin et al., 1976), uniform limit theorem (Rudin et al., 1976), and the extreme value theorem. Different from the definitions of the state-robust totally optimal agent policy and robust total Nash equilibrium, the worst-case expected state value objective does not require the optimality condition to hold for all states. Agents won't get stuck in trade-offs between different states, therefore, we can find a robust agent policy to maximize the worst-case expected state value for the SAMG problem.

The robust agent policy, while motivated in specific instances of non-existence theorems, is designed to offer broader applicability. The significance of the new solution concept lies in providing alternative solutions where traditional methods may falter or be inapplicable. This versatility is crucial in advancing the field, particularly in complex scenarios where standard solutions are inadequate.

**Multi-Agent Adversarial Actor-Critic (RMA3C) Algorithm**   In general, it is challenging to develop algorithms that compute optimal or equilibrium policies for MARL under uncertainties (Zhang et al., 2020b; 2021). We design a RMA3C Algorithm based on our theoretical analysis above. Each agent has one critic network, one actor network $\pi^i$ and one adversary network $\chi^i$. The critic $Q$ takes in the true global state and global action during the training process. It returns a $Q$-value denoting the total expected return given $s$ and $a$. We use Gradient Descent Ascent (GDA) optimizer (Lin et al., 2020b) to update parameters for each agent's actor network and adversary network for the maximin problem $\max_\pi \min_\chi \sum_{s_0} \Pr(s_0) V_{\pi,\chi}(s_0)$ in Theorem 4.10. A detailed introduction for the RMA3C and pseudocode is included in Appendix D.

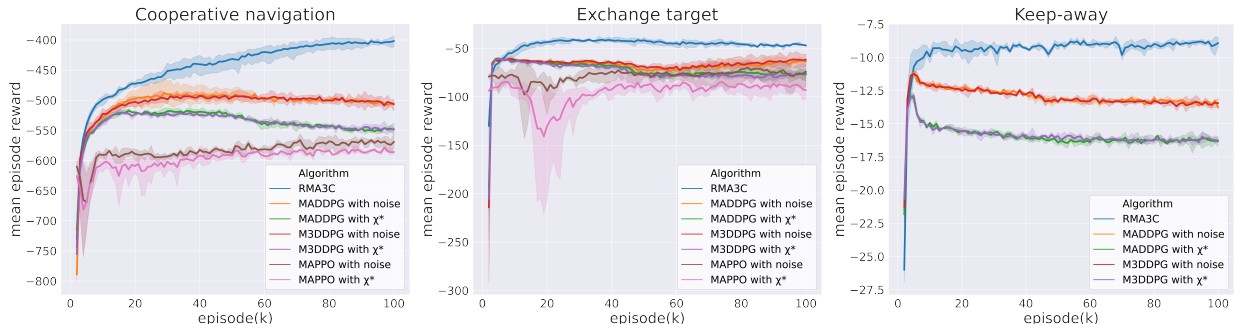

Figure 4: Our RMA3C algorithm compared with several baseline algorithms in training. The results show that our RMA3C algorithm outperforms the baselines, achieving higher mean episode rewards and greater robustness to state perturbations. The baselines were trained under either random state perturbations or a well-trained adversary policy $\chi^*$ (adversaries that are trained for the maximum training episodes in RMA3C). Overall, our RMA3C algorithm achieved up to 58.46% higher mean episode rewards than the baselines.

# 5   Experiments

To demonstrate the effectiveness of our algorithm, we utilize the multi-agent particle environments developed in Lowe et al. (2017) which consist of multiple agents and landmarks in a 2D world. The host machine adopted in our experiments is a server configured with AMD Ryzen Threadripper 2990WX 32-core processors and four Quadro RTX 6000 GPUs. Our experiments are performed on Python 3.5.4, Gym 0.10.5, Numpy 1.14.5, Tensorflow 1.8.0, and CUDA 9.0. In our experiments, we consider the set of admissible perturbed

Table 1: Mean episode reward of 2000 episodes during testing. Our RMA3C policy achieves up to 46.56% higher mean episode rewards than the baselines with random state perturbations $\mathcal{N}$.

| Environment | CN | ET | KA | PD |
|---|---|---|---|---|
| MA Lowe et al. (2017) | $-388.59 \pm 60.72$ | $-45.79 \pm 23.50$ | -8.80 $\pm$ 5.07 | 3.03 $\pm$ 0.67 |
| M3 Li et al. (2019) | -390.94 $\pm$ 59.83 | -39.55 $\pm$ 20.53 | -8.54 $\pm$ 5.04 | 2.12 $\pm$ 1.04 |
| MP Yu et al. (2022) | -381.70 $\pm$ 54.06 | -37.62 $\pm$ 18.94 | - | - |
| MADDPG (MA) w/ $\mathcal{N}$ | -487.67 $\pm$ 72.28 | -55.79 $\pm$ 26.78 | -11.21 $\pm$ 6.82 | 1.24 $\pm$ **0.47** |
| M3DDPG (M3) w/ $\mathcal{N}$ | -478.96 $\pm$ 70.27 | -54.40 $\pm$ 26.64 | -11.28 $\pm$ 6.71 | 1.30 $\pm$ 0.58 |
| MAPPO (MP) w/ $\mathcal{N}$ | -523.83 $\pm$ 78.45 | -86.51 $\pm$ 30.86 | - | - |
| RMA3C w/ $\mathcal{N}$ (ours) | **-390.20 $\pm$ 64.82** | **-46.23 $\pm$ 24.76** | **-9.02 $\pm$5.87** | **2.48** $\pm$ 1.26 |

state for agent $i$ at state $s$ as an $\ell_\infty$ norm ball around $s$: $\mathcal{P}_s^i := \{\rho^i \in \mathcal{S} : \|\rho^i - s\|_\infty \leq d\}$ where $d$ is a radius denoting the perturbation budget. In implementation, the adversary network takes in the true state $s$ and learns a state perturbation vector $\Delta^i$ and we project $s + \Delta^i$ to $\mathcal{P}_s^i$. The environments used in our experiments include cooperative navigation (CN), exchange target (ET), keep-away (KA), and physical deception (PD). A detailed introduction to these environments can be found in Appendix E. All hyperparameters used in our experiments for RMA3C and the baselines are listed in Appendix E, along with additional implementation details and experiment results.

## 5.1 Baselines

In our experiment, we have a total of 9 baselines: MADDPG Lowe et al. (2017), M3DDPG Li et al. (2019), MAPPO Yu et al. (2022), as well as versions of these algorithms with random and well-trained adversarial state perturbations. Detailed explanation of these baselines can be found in Appendix E.2. To evaluate robustness under state uncertainty, we add state noise to MADDPG, M3DDPG, and MAPPO produced by a truncated normal distribution $\mathcal{N}(0, \lambda, u, l)$ where $\lambda$ is the uncertainty level, $u$ and $l$ are the upper and lower bounds to ensure noise's compactness. This simulates adversaries selecting random state perturbations. In contrast, our RMA3C algorithm trains agents under adversaries that try to minimize the agents' total expected return. We save the well-trained adversaries $\chi^*$ for each scenario in RMA3C to represent the optimal state perturbation adversaries. The well-trained adversaries are the adversary policies trained in the RMA3C algorithm when the algorithm reaches the maximum training episodes (100k episodes). We then use these adversaries to perturb the states for MADDPG, M3DDPG, and MAPPO to train and test their robustness under adversarial state perturbations. Because MAPPO provided in Yu et al. (2022) only works in fully cooperative tasks, we only report its results in cooperative navigation and exchange target. For both training and testing, we report statistics that are averaged across 10 runs in each scenario and algorithm.

## 5.2 Comparison Results

**Training Comparison Under different Perturbations** We compare our RMA3C algorithm with baselines during the training process to demonstrate its superiority in terms of mean episode rewards under different state perturbations as shown in Fig. 4. As RMA3C has a built-in adversary to perturb states, we do not train it under random state perturbations. In comparison to other baselines with different state perturbations, RMA3C consistently achieved higher mean episode rewards, demonstrating its robustness under varying state perturbations. Furthermore, when comparing each baseline with random state perturbations to the same baseline with the well-trained adversary policy $\chi^*$, we can see that the adversary policy trained by RMA3C is more effective than random state perturbations. This is because $\chi^*$ is designed to intentionally select state perturbations that minimize the agents' total expected return. The mean episode rewards of the last 1000 episodes during training are shown in the table in Appendix E.5. Our RMA3C algorithm achieved up to 58.46% higher mean episode rewards than the baselines under different state perturbations.

**Training Comparison With More Agents** Our RMA3C algorithm is compared with baselines in the cooperative navigation scenario with an increasing number of agents. As shown in Fig.4, the original cooperative navigation environment has 3 agents and our RMA3C algorithm outperforms the baselines in terms of mean episode rewards. In Fig.5(a), we present the results of training with 4 agents, where our

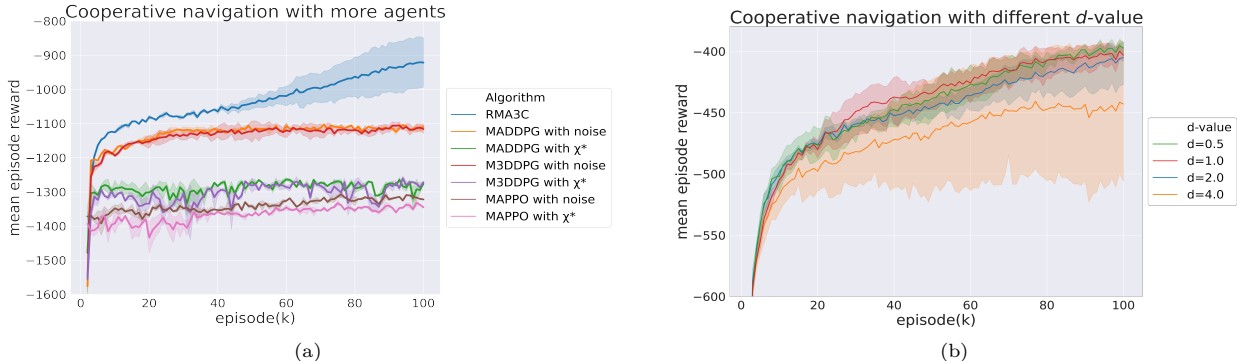

Figure 5: 5(a): Our RMA3C algorithm continues to achieve higher mean episode rewards, even with an increasing number of agents in the environment. 5(b):Our RMA3C algorithm is trained in the cooperative navigation environment with different perturbation budgets $d$. When $d$ increases, adversaries get more advantage, and may further decrease agents' total expected return.

RMA3C algorithm still surpasses the baselines. We include the training results with 6 agents in Appendix E.6. Our RMA3C algorithm continues to achieve higher mean episode rewards, even with an increasing number of agents in the environment.

Table 2: Our RMA3C policy achieves up to 54.02% higher mean episode reward than the baselines with well-trained $\chi^*$.

| Environment | CN | ET | KA | PD |
|---|---|---|---|---|
| MADDPG (MA) w/$\chi^*$ | -537.56 ± 72.28 | -71.65 ± 42.50 | -14.72 ± 5.44 | -0.95 ± 1.32 |
| M3DDPG (M3) w/$\chi^*$ | -515.85 ± 74.58 | -70.68 ± 41.54 | -13.51 ± **5.30** | -0.70 ± 0.96 |
| MAPPO (MP) w/$\chi^*$ | -572.39 ± 79.34 | -109.26 ± 47.97 | - | - |
| RMA3C w/$\chi^*$ (ours) | **-400.82 ± 62.59** | **-50.23 ± 26.97** | **-9.64** ± 5.31 | **1.23 ± 0.82** |

**Training Comparison With Different Perturbation Budgets** We compare our algorithm with baselines in the cooperative navigation scenario with varying levels of perturbation budgets $d$. We consider the set of admissible perturbed state for agent $i$ at state $s$ as an $\ell_\infty$ norm ball around $s$: $\mathcal{P}_s^i := \{\rho^i \in \mathcal{S} : \|\rho^i - s\|_\infty \leq d\}$ where $d$ is a radius denoting the perturbation budget. As shown in Fig. 5(b), when $d$ increases, adversaries have greater freedom to perturb the state within a larger admissible perturbed state set. As $d$ increases, adversaries get more powerful and lead to a decrease in agents' total expected return.

**Testing Comparison in different Environments** Our RMA3C algorithm is tested in different environments to demonstrate its robustness under state perturbations. As shown in Table 1, the mean episode rewards are averaged across 2000 episodes and 10 test runs in each environment. The results of MADDPG, M3DDPG, and MAPPO, which are not designed to handle state perturbations, are shown as a reference for the no state perturbation scenario. These algorithms perform poorly when random state perturbations are introduced, indicating the need for an algorithm that can handle state perturbations. As seen in Table 2, the RMA3C policy achieves up to 46.56% higher mean episode rewards than the baselines in environments with random state perturbations. Additionally, we also test the learned policies using different algorithms in environments with well-trained adversary policies $\chi^*$ to perturb states. The results indicate that the RMA3C policy achieves up to 54.02% higher mean episode reward than the baselines with well-trained adversarial state perturbations. Overall, these tests demonstrate that the RMA3C algorithm achieves higher robustness in different environments with state perturbations.

# 6 Conclusion

In this work, we propose a State-Adversarial Markov Game (SAMG) and investigate the fundamental properties of robust MARL under adversarial state perturbations. We prove that the widely used solution

concepts such as optimal agent policy and robust Nash equilibrium do not always exist for SAMGs. Instead, we consider a new solution concept (the robust agent policy) to maximize the worst-case expected state value and prove its existence. This is the primary theoretical contribution of our work. Additionally, we also propose a RMA3C algorithm to find a robust policy for MARL agents under state perturbations. Our numerical experiments demonstrate that the RMA3C algorithm improves the robustness of the trained policies against both random and adversarial state perturbations. Some discussions and future directions are provided in Appendix F.

### Acknowledgments

Songyang Han, Sanbao Su, Sihong He, and Fei Miao are supported by the National Science Foundation under Grants CNS-1952096, and CNS-2047354 grants. Haizhao Yang was partially supported by the US National Science Foundation under awards DMS-2244988, DMS-2206333, and the Office of Naval Research Award N00014-23-1-2007.

Shaofeng Zou is supported by the National Science Foundation under Grants CCF-2106560, and CCF-2007783. This material is based upon work supported under the AI Research Institutes program by National Science Foundation and the Institute of Education Sciences, U.S. Department of Education through Award # 2229873 - National AI Institute for Exceptional Education. Any opinions, findings and conclusions or recommendations expressed in this material are those of the author(s) and do not necessarily reflect the views of the National Science Foundation, the Institute of Education Sciences, or the U.S. Department of Education.

We extend our thanks to Peter Stone and Dustin Morrill in Sony AI for their assistance in proofreading this paper and for their insightful suggestions that have significantly enhanced the quality of this work. Their careful attention to detail and valuable feedback were greatly appreciated.

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

# A    Comparison with Dec-POMDP and Markov Games

## A.1    Comparison with Dec-POMDP

Our SAMG problem cannot be solved by the existing work in the Decentralized Partially Observable Markov Decision Process (Dec-POMDP) (Oliehoek et al., 2016). In contrast, the policy in our problem needs to be robust under a set of admissible perturbed states. The adversary aims to find the worst-case state perturbation policy $\chi$ to minimize the MARL agents' total expected return. In the following proposition, we show that under certain additional conditions our proposed SAMG problem becomes a Dec-POMDP problem.

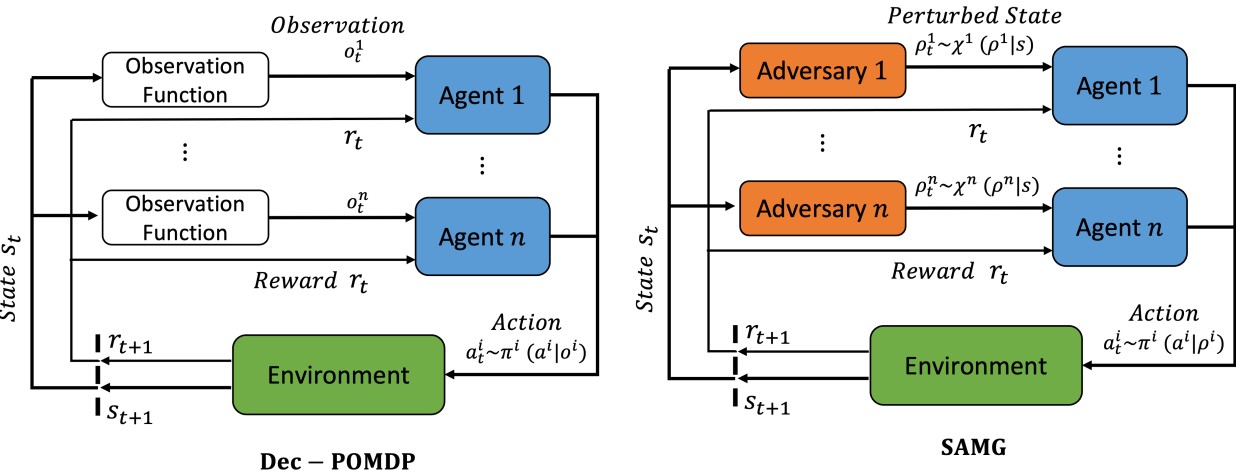

Figure 6: Comparison between Dec-POMDP and SAMG. In Dec-POMDP, the observation probability function is fixed, and it will not change according to the change of the agent policy. However, in SAMG the adversary policy is not a fixed policy, it may change according to the agents' policies and always select the worst-case state perturbation for agents.

**Proposition 3.2.** *When the adversary policy is a fixed policy, the SAMG problem becomes a Dec-POMDP (Oliehoek et al., 2016).*

*Proof.* When the adversary policy $\chi$ is a fixed policy, an SAMG $(\mathcal{N}, \mathcal{S}, \mathcal{A}, r, \mathcal{P}_s, p, \gamma, \Pr(s_0))$ becomes a Dec-POMDP $(\mathcal{N}, \mathcal{S}, \mathcal{A}, r, \mathcal{O}, O, p, \gamma, \Pr(s_0))$. The agent set $\mathcal{N} = \{1, ..., n\}$. The global joint state is $s \in \mathcal{S}$. Each agent $i$ is associated with an action $a^i \in \mathcal{A}^i$. The global joint action is $a = (a^1, ..., a^n) \in \mathcal{A}$, $\mathcal{A} := \mathcal{A}^1 \times \cdots \times \mathcal{A}^n$. All agents share a stage-wise reward function $r : \mathcal{S} \times \mathcal{A} \to \mathbb{R}$. The state transition function is $p : \mathcal{S} \times \mathcal{A} \to \Delta(\mathcal{S})$, where $\Delta(\mathcal{S})$ is a probability simplex denoting the set of all possible probability measures on $\mathcal{S}$. The state transits from the true state to the next state. The discount factor is $\gamma$. The joint observation set $\mathcal{O}$ is the same as the joint state set $\mathcal{S}$. The observation probability function $O(o|s) = \chi(o|s)$ for any $o \in \mathcal{P}_s$ and $O(o|s) = 0$ for any $o \notin \mathcal{P}_s$, where $o$ is the observation given the state $s$. The $\Pr(s_0)$ is the probability distribution of the initial state. $\square$

In Dec-POMDP, the observation probability function is fixed, and it will not change according to the change of the agent policy. However, in SAMG the adversary policy is not a fixed policy, it may change according to the agents' policies and always select the worst-case state perturbation for agents. In contrast to Dec-POMDP, the adversary's policy $\chi$ is chosen to minimize the total expected return of the agents in our problem. Additionally, in Dec-POMDP the agents do not have access to the true state $s$, whereas in our problem, the adversaries are aware of the true state and can use it to select perturbed states.

## A.2 SAMG cannot be solved by Dec-POMDP: Two-Agent Two-State Game Example

We use a two-agent two-state game to show the difference between Dec-POMDP and SAMG. Consider a game with two agents $\mathcal{N} = \{1, 2\}$ and two states $\mathcal{S} = \{s_1, s_2\}$ as shown in Fig. 7. Each agent has two actions $\mathcal{A}^1 = \mathcal{A}^2 = \{a_1, a_2\}$. The transition probabilities are defined below.

$$
\begin{aligned}
p(s' = s_1 | s = s_1, a^1 \neq a^2) &= 1, \\
p(s' = s_2 | s = s_1, a^1 = a^2) &= 1, \\
p(s' = s_2 | s = s_2, a^1 \neq a^2) &= 1, \\
p(s' = s_1 | s = s_2, a^1 = a^2) &= 1.
\end{aligned}
\tag{6}
$$

Specifically, $a^1 = a^2$ includes two cases: $a^1 = a^2 = a_1$ or $a^1 = a^2 = a_2$. Similarly, $a^1 \neq a^2$ includes two cases: $a^1 = a_1, a^2 = a_2$ or $a^1 = a_2, a^2 = a_1$.

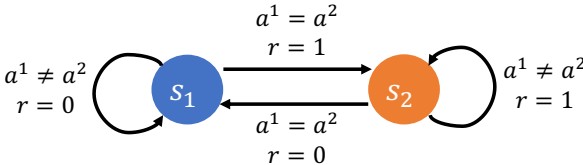

Figure 7: A two-agent two-state game example. Agents get reward 1 at state $s_1$ if they choose the same action. Agents get reward 1 at state $s_2$ if they choose different actions.

Two agents share the same reward function:

$$
r(s, a^1, a^2) = \begin{cases}
1, & a^1 = a^2, \text{and } s = s_1, \\
0, & a^1 \neq a^2, \text{and } s = s_1, \\
0, & a^1 = a^2, \text{and } s = s_2, \\
1, & a^1 \neq a^2, \text{and } s = s_2.
\end{cases}
\tag{7}
$$

In a SAMG, each agent is associated with an adversary to perturb its knowledge or observation of the true state. For the power of the adversary, we allow the adversary to perturb any state to the other state:

$$
\mathcal{P}_s^1 = \mathcal{P}_s^2 = \{s_1, s_2\}.
\tag{8}
$$

We use $\gamma = 0.99$ as the discount factor. Agents want to find a policy $\pi$ to maximize their total expected return while adversaries want to find a policy $\chi$ to minimize agents' total expected return.

**This problem cannot be formulated as a Dec-POMDP** Consider one agent policy where both agents select the same action in $s_1$ and select different actions in $s_2$: $\pi^1(a_1|s_1) = \pi^1(a_1|s_2) = \pi^2(a_1|s_1) = \pi^2(a_2|s_2) = 1$. When there is no adversary, agents keep receiving rewards. The values for each state are $\tilde{V}(s_1) = \tilde{V}(s_2) = \frac{1}{1-\gamma} = 100$. Because agents share the same reward function, they also share the same values for each state. However, this policy receives $V(s_1) = V(s_2) = 0$ when agents are facing the worst-case adversaries $\chi^i(s_1|s_2) = \chi^i(s_2|s_1) = 1$ for $i = 1, 2$ and always taking the wrong actions with 0 reward.

If the adversary policy is fixed at $\chi^i(s_1|s_2) = \chi^i(s_2|s_1) = 1$ for $i = 1, 2$, this problem becomes a Dec-POMDP with the observation space $\mathcal{O} = \{o_1 = s_1, o_2 = s_2\}$. The observation function is $o^i(o_1|s_2) = o^i(o_2|s_1) = 1$ for $i = 1, 2$. The agent policy is $\pi^1(a_1|o_1) = \pi^1(a_1|o_2) = \pi^2(a_1|o_1) = \pi^2(a_2|o_2) = 1$.

However, when we consider a different agent policy where both agents select the same action in $s_2$ and select different actions in $s_1$: $\pi^1(a_1|s_2) = \pi^1(a_1|s_1) = \pi^2(a_1|s_2) = \pi^2(a_2|s_1) = 1$, agents keep receiving 0 rewards even when the adversary does nothing. For the new agent policy, the worst-case adversary policy is $\chi^i(s_1|s_1) = \chi^i(s_2|s_2) = 1$ for $i = 1, 2$. The corresponding observation function for the new adversary policy is $o^i(o_1|s_1) = o^i(o_2|s_2) = 1$ for $i = 1, 2$, which is completely different from the previous observation functions. Because the observation function in Dec-POMDP won't change according to agents' policies, therefore, the SAMG problem cannot be formulated by Dec-POMDP when adversary policy is not fixed.

**Under different observation functions, Dec-POMDP can lead to contradictory agent policies.** Besides the analysis of why this problem cannot be formulated as a Dec-POMDP, we also demonstrate that Dec-POMDPs fail to solve this problem from a different perspective.

Let's consider a Dec-POMDP with the observation space $\mathcal{O} = \{o_1 = s_1, o_2 = s_2\}$. The observation function is defined as $o^i(o_1|s_2) = o^i(o_2|s_1) = 1$ for $i = 1, 2$. In this scenario, the optimal agent policy is to select the same action in response to $o_2$ and choose different actions for $o_1$. Agents keep receiving rewards based on this policy.

Now let's consider another Dec-POMDP with the observation space $\mathcal{O} = \{o_1 = s_1, o_2 = s_2\}$. The observation function is defined as $o^i(o_1|s_1) = o^i(o_2|s_2) = 1$ for $i = 1, 2$. In this case, the optimal agent policy is to select the same action in response to $o_1$ and choose different actions for $o_2$. Agents keep receiving rewards based on this policy. However, the new optimal agent policy contradicts the previous one.

By comparing these two Dec-POMDPs with different observation functions, we observe that Dec-POMDPs can yield different agent policies based on different observation functions. This implies that Dec-POMDPs do not address the problem of selecting an agent policy when the observation function is determined by an adversary.

Furthermore, we will reanalyze this problem and demonstrate how a SAMG can solve this two-agent two-state game in Appendix B and C. The SAMG formulation addresses this problem by selecting the agent policy against the worst-case observation function.

### A.3  Comparison with Markov Games

Under a specific condition, when the adversary policy $\chi$ is a bijective mapping from $\mathcal{S}$ to $\mathcal{S}$, the SAMG problem is equivalent to a Markov game, as demonstrated in the following proposition. This proposition illustrates the relationship between a SAMG and a Markov game with a particular form of state perturbation.

When $\chi$ is a bijective mapping from $\mathcal{S}$ to $\mathcal{S}$, the adversary policy follows $\chi(\rho|s) = 1$ selecting the perturbed state $\rho$ for the true state $s$ with probability 1. Let us use the notation $\chi(s) = \rho$ for this special case.

**Proposition 3.3.** *When the adversary policy is a fixed bijective mapping from $\mathcal{S}$ to $\mathcal{S}$, the SAMG problem becomes a Markov game.*

*Proof.* When the adversary policy $\chi$ is a fixed bijective mapping from $\mathcal{S}$ to $\mathcal{S}$, an SAMG problem $(\mathcal{N}, \mathcal{S}, \mathcal{A}, r, \mathcal{P}_s, p, \gamma, \Pr(s_0))$ becomes a Markov game $(\mathcal{N}_{new}, \mathcal{S}_{new}, \mathcal{A}_{new}, r^i_{new}, p_{new}, \gamma, \Pr(s_{new,0}))$ that is constructed as follows:

Taking $s_{new} = \rho = \chi(s)$ as the new state, the new global joint state set is $\mathcal{S}_{new} := \mathcal{S}$. The global joint action set $\mathcal{A}_{new} = \mathcal{A} = \mathcal{A}^1 \times \cdots \times \mathcal{A}^n$ and the agent set $\mathcal{N}_{new} = \mathcal{N}$ stay the same.

We can construct a new reward function $r^i_{new} : \mathcal{S}_{new} \times \mathcal{A}_{new} \to \mathbb{R}$ for each agent $i$ as

$$r^i_{new}(s_{new} = \chi(s), a_{new} = a) = r(s, a), \tag{9}$$

and a new state transition function $p_{new} : \mathcal{S}_{new} \times \mathcal{A}_{new} \to \Delta(\mathcal{S}_{new})$ defined as

$$p_{new}(\rho' = \chi(s')|\rho = \chi(s), a) = p(s'|s, a). \tag{10}$$

The new probability of the initial state is

$$\Pr(s_{new,0} = \chi(s_0)) = \Pr(s_0). \tag{11}$$

Each agent uses a policy $\pi^i_{new} : \mathcal{S}_{new} \to \Delta(\mathcal{A}^i)$ to choose an action based on the new state. Hence, the SAMG problem becomes a Markov game. $\square$

If the adversary's policy $\chi$ is a fixed bijective mapping from $\mathcal{S}$ to $\mathcal{S}$, the new global joint state set $\mathcal{S}_{new}$ is a perturbation of $\mathcal{S}$ and each state is assigned a new "label" by the adversary. Under this condition, the SAMG is equivalent to a Markov game.

# B  Optimal Adversary Policy and Optimal Agent Policy

In this section, we analyze the existence of the optimal adversary policy and the optimal agent policy. We will utilize the two-agent two-state game introduced in Appendix A. For completeness, let us revisit this game with two agents $\mathcal{N} = \{1, 2\}$ and two states $\mathcal{S} = \{s_1, s_2\}$ as shown in Fig. 8. Each agent has two actions $\mathcal{A}^1 = \mathcal{A}^2 = \{a_1, a_2\}$. The transition probabilities are defined below.

$$
\begin{aligned}
p(s' = s_1 | s = s_1, a^1 \neq a^2) &= 1, \\
p(s' = s_2 | s = s_1, a^1 = a^2) &= 1, \\
p(s' = s_2 | s = s_2, a^1 \neq a^2) &= 1, \\
p(s' = s_1 | s = s_2, a^1 = a^2) &= 1.
\end{aligned}
\tag{12}
$$

Specifically, $a^1 = a^2$ includes two cases: $a^1 = a^2 = a_1$ or $a^1 = a^2 = a_2$. Similarly, $a^1 \neq a^2$ includes two cases: $a^1 = a_1, a^2 = a_2$ or $a^1 = a_2, a^2 = a_1$.

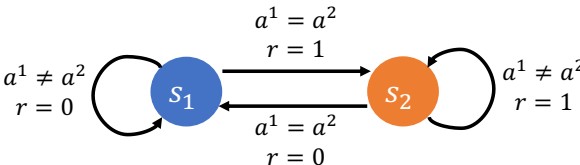

Figure 8: A two-agent two-state game example. Agents get reward 1 at state $s_1$ if they choose the same action. Agents get reward 1 at state $s_2$ if they choose different actions. This example was used in Appendix A to show the difference between Dec-POMDP and SAMG. We will revisit this game in Appendix B to discuss optimal adversary policy and optimal agent policy.

Two agents share the same reward function:

$$
r(s, a^1, a^2) = \begin{cases}
1, & a^1 = a^2, \text{and } s = s_1, \\
0, & a^1 \neq a^2, \text{and } s = s_1, \\
0, & a^1 = a^2, \text{and } s = s_2, \\
1, & a^1 \neq a^2, \text{and } s = s_2.
\end{cases}
\tag{13}
$$

In a SAMG, each agent is associated with an adversary to perturb its knowledge or observation of the true state. For the power of the adversary, we allow the adversary to perturb any state to the other state:

$$
\mathcal{P}_s^1 = \mathcal{P}_s^2 = \{s_1, s_2\}.
\tag{14}
$$

We use $\gamma = 0.99$ as the discount factor. Agents want to find a policy $\pi$ to maximize their total expected return while adversaries want to find a policy $\chi$ to minimize agents' total expected return.

## B.1  Optimal Agent Policy Without Adversaries

When there is no adversary, the optimal policy for agents is to choose the same action in $s_1$ and choose different actions in $s_2$. One example is $\pi^1(a_1|s_1) = \pi^1(a_1|s_2) = \pi^2(a_1|s_1) = \pi^2(a_2|s_2) = 1$. The agents keep receiving rewards. The values for each state are $\tilde{V}(s_1) = \tilde{V}(s_2) = \frac{1}{1-\gamma} = 100$. Because agents share the same reward function, they also share the same values for each state. However, this policy receives $V(s_1) = V(s_2) = 0$ when agents are facing adversaries $\chi^i(s_1|s_2) = \chi^i(s_2|s_1) = 1$ for $i = 1, 2$ and always taking the wrong actions with 0 reward.

## B.2  A Stochastic Policy With Adversaries

We consider a stochastic policy $\pi^1(a_1|s_1) = \pi^1(a_1|s_2) = \pi^2(a_1|s_1) = \pi^2(a_2|s_2) = 0.5$. Under this policy, the probabilities of taking the same or different actions are the same for each state $\Pr(a^1 = a^2 \mid s_1) = \Pr(a^1 \neq$

$a^2 \mid s_1) = \Pr(a^1 = a^2 \mid s_2) = \Pr(a^1 \neq a^2 \mid s_2) = 0.5$. Agents randomly stay or transit in each state and receive a positive reward with a 50% probability. The adversary has no power under this policy because $\pi$ is the same for both states. The values for each state are $V(s_1) = V(s_2) = \tilde{V}(s_1) = \tilde{V}(s_2) = \frac{0.5}{1-\gamma} = 50$.

### B.3 Deterministic Policies With Adversaries

Since each agent has two actions for each state, there are in total $2^4 = 16$ possible deterministic policies for the two-agent two-state game example. All possible deterministic policies can be classified into three cases: (1) If agents select the same action in one state $s_i$ and select different actions in the other state $s_j$, then we always have $V(s_1) = V(s_2) = 0$. This is because adversaries can always use $\chi^k(s_1|s_j) = \chi^k(s_2|s_i) = 1$ for $k = 1, 2$ such that agents always receive a 0 reward. (2) If agents always select different actions in both states, then $V(s_1) = 0, V(s_2) = 100$. This is because agents never transit to the other state and keep receiving the same reward. (3) If agents always select the same action in both states, then $V(s_1) = \frac{1}{1-\gamma^2} \approx 50.25, V(s_2) = \frac{\gamma}{1-\gamma^2} \approx 49.75$. This is because agents circulate through both states and adversaries have no power to change it.

### B.4 Optimal Adversary Policy

In this section, we examine optimal policies for both the adversary and the agent in a State-Adversarial Markov Game (SAMG). The following proposition demonstrates the existence of an optimal adversary in an SAMG.

**Proposition 4.1 (Existence of Optimal Adversary Policy).** *Given an SAMG, for any given agent policy, there exists an optimal adversary policy.*

*Proof.* We prove this by constructing an MDP $M = (\mathcal{S}, \hat{\mathcal{A}}, \hat{r}, \hat{p}, \gamma)$ such that an optimal policy of $M$ is an optimal adversary policy $\chi^*$ for the SAMG given the fixed $\pi$. In the MDP $M$, we take all adversaries as a joint adversary agent. The joint adversary learns a policy $\chi$ to find a joint perturbed state given the current true state. The action space $\hat{\mathcal{A}} = \mathcal{S} \times \mathcal{S} \times \cdots \times \mathcal{S}$. Note that the joint admissible perturbed state set in Definition 3.1 $\mathcal{P}_s \subseteq \hat{\mathcal{A}}$.

The reward function $\hat{r}$ is defined as:

$$\hat{r}(s, \hat{a}) = -\sum_{a \in \mathcal{A}} \pi(a|\hat{a})r(s,a) \text{ for } \hat{a} \in \mathcal{P}_s. \tag{15}$$

The transition probability $\hat{p}$ is defined as

$$\hat{p}(s'|s, \hat{a}) = \sum_{a \in \mathcal{A}} \pi(a|\hat{a})p(s'|s,a) \text{ for } \hat{a} \in \mathcal{P}_s. \tag{16}$$

The reward function is defined based on the intuition that when the agent receives $r$ given $s, a$, the reward of the adversary is the negative of the agent reward, that is to say, $\hat{r} = -r$. Considering that $r(s,a) = \mathbb{E}[R|s,a] = -\mathbb{E}[\hat{R}|s,a]$,

$$\begin{aligned}
\hat{r}(s, \hat{a}) &= \mathbb{E}[\hat{R}|s, \hat{a}] \\
&= \sum_{\hat{R}} \hat{R} \sum_{a \in \mathcal{A}} \Pr[\hat{R}|s, a]\pi(a|\hat{a}) \\
&= \sum_{a \in \mathcal{A}} \left[ \sum_{\hat{R}} \hat{R} \Pr[\hat{R}|s, a] \right] \pi(a|\hat{a}) \\
&= \sum_{a \in \mathcal{A}} \mathbb{E}[\hat{R}|s, a]\pi(a|\hat{a}) \\
&= -\sum_{a \in \mathcal{A}} \mathbb{E}[R|s, a]\pi(a|\hat{a})
\end{aligned}$$

$$= -\sum_{a \in \mathcal{A}} r(s,a)\pi(a|\hat{a}). \tag{17}$$

Based on the properties of MDP (Sutton et al., 1998; Puterman, 2014), we know that the MDP $M$ has an optimal policy $\chi^*$ that satisfies $\hat{V}_{\pi,\chi^*}(s) \geq \hat{V}_{\pi,\chi}(s)$ for all $s$ and all $\chi$, where $\hat{V}_{\pi,\chi}$ is the state value function of the MDP $M$.

The Bellman equation for the MDP $M$ is

$$\hat{V}_{\pi,\chi}(s) = \sum_{\hat{a} \in \mathcal{P}_s} \chi(\hat{a}|s) \left( \hat{r} + \gamma \sum_{s' \in \mathcal{S}} \hat{p}(s'|s,\hat{a})\hat{V}_{\pi,\chi}(s') \right)$$

$$= \sum_{\hat{a} \in \mathcal{P}_s} \chi(\hat{a}|s) \sum_{a \in \mathcal{A}} \pi(a|\hat{a}) \left( -r + \gamma \sum_{s' \in \mathcal{S}} p(s'|s,a)\hat{V}_{\pi,\chi}(s') \right). \tag{18}$$

By multiplying $-1$ on both sides, we have

$$(-\hat{V}_{\pi,\chi}(s)) = \sum_{\hat{a} \in \mathcal{P}_s} \chi(\hat{a}|s) \sum_{a \in \mathcal{A}} \pi(a|\hat{a})$$

$$\left[ r + \gamma \sum_{s' \in \mathcal{S}} p(s'|s,a)(-\hat{V}_{\pi,\chi}(s')) \right]. \tag{19}$$

On the other side, for the SAMG, we have the Bellman equation for any fixed policies $\pi$ and $\chi$ as

$$V_{\pi,\chi}(s) = \sum_{\rho \in \mathcal{P}_s} \chi(\rho|s) \sum_{a \in \mathcal{A}} \pi(a|\rho)$$

$$\left( r + \gamma \sum_{s' \in \mathcal{S}} p(s'|s,a)V_{\pi,\chi}(s') \right). \tag{20}$$

When $\pi$ and $\chi$ are fixed, they can be taken together as a single policy, and the existing results from Dec-POMDP can be directly applied. Comparing Eq. (20) and (19), we know that $V_{\pi,\chi}(s) = (-\hat{V}_{\pi,\chi}(s))$.

An optimal adversary policy $\chi^*$ for the MDP $M$ satisfies $\hat{V}_{\pi,\chi^*}(s) \geq \hat{V}_{\pi,\chi}(s)$ for any $s$ and any $\chi$. Therefore, $\chi^*$ also satisfies $V_{\pi,\chi^*}(s) \leq V_{\pi,\chi}(s)$ for any $s$ and any $\chi$, and an optimal policy of the MDP $M$ is an optimal adversary policy for the SAMG given the fixed $\pi$.

$\square$

### B.5 Optimal Agent Policy With Adversaries

We have established the existence of an optimal adversary in SAMGs. Next, we consider a state-robust totally optimal agent policy under this optimal adversary. The following proposition demonstrates that a deterministic agent policy is not always superior to a stochastic policy in SAMGs.

**Proposition B.1.** *There exists an SAMG and some stochastic policy $\pi$ such that we cannot find a better deterministic policy $\pi'$ satisfying $\bar{V}_{\pi'}(s) \geq \bar{V}_\pi(s)$ for all $s \in \mathcal{S}$.*

*Proof.* We prove this theorem by giving a counter-example where no deterministic policy is better than a stochastic policy. As shown in the two-agent two-state game example in Fig. 8, all 16 deterministic policies are no better than the stochastic policy $\pi^1(a_1|s_1) = \pi^1(a_1|s_2) = \pi^2(a_1|s_1) = \pi^2(a_2|s_2) = 0.5$. $\square$

Finally, we show a state-robust totally optimal agent policy $\pi^*$ does not always exist such that $\bar{V}_{\pi^*}(s) \geq \bar{V}_\pi(s)$ for any $\pi$ and all $s \in \mathcal{S}$ in SAMGs in the following theorem.

**Theorem 4.3 (Non-existence of State-robust Totally Optimal Agent Policy).** *A state-robust totally optimal agent policy does not always exist for SAMGs.*

*Proof.* We prove this theorem by showing that the two-agent two-state game in Fig. 8 does not have an optimal policy. We first show that the policy $\pi_1 : \pi^1(a_1|s_1) = \pi^1(a_1|s_2) = \pi^2(a_2|s_1) = \pi^2(a_2|s_2) = 1$ is not an optimal policy. Because agents always select different actions in both states, agents always stay in the same state and adversaries have no power to change it. The values for each state are $\bar{V}_{\pi_1}(s_1) = 0, \bar{V}_{\pi_1}(s_2) = 100$. Now we consider the stochastic policy $\pi_2 : \pi^1(a_1|s_1) = \pi^1(a_1|s_2) = \pi^2(a_1|s_1) = \pi^2(a_2|s_2) = 0.5$. The values for each state are $\bar{V}_{\pi_2}(s_1) = \bar{V}_{\pi_2}(s_2) = 50$. Because $\bar{V}_{\pi_2}(s_1) > \bar{V}_{\pi_1}(s_1)$, the policy $\pi_1$ is not an optimal policy for agents.

If there exists an optimal policy $\pi^*$, then it must be better than $\pi_1$ and have $\bar{V}_{\pi^*}(s_1) > 0, \bar{V}_{\pi^*}(s_2) = 100$. In order to have $\bar{V}_{\pi^*}(s_2) = 100$, agents must select different actions in $s_2$ and keep receiving the positive rewards from each step. In order to have $\bar{V}_{\pi^*}(s_1) > 0$, agents must have a chance to select the same action in $s_1$, i.e., $\Pr(a^1 = a^2 \mid s_1) > 0$. However, if $\Pr(a^1 = a^2 \mid s_1) > 0$, then adversaries can have $\chi^i(s_1|s_2) > 0$ for $i = 1, 2$ to perturb the state $s_2$ to $s_1$ and reduce $\bar{V}_{\pi^*}(s_2)$. Therefore, no policy can do better than $\pi_1$ and since $\pi_1$ is not an optimal policy, there is no optimal policy for agents. $\qquad\square$

In the comparison of $\pi_1$ and $\pi_2$ in the above proof, it is apparent that it is not always possible to maximize the state value of all states and that trade-offs may need to be made among different states. Using the traditional definition of an optimal policy, it is not possible to determine which policy, $\pi_1$ or $\pi_2$, is better. However, if we use the worst-case expected state value concept from Definition 4.8 and assume that the initial state is always $s_2$, then we can conclude that $\pi_1$ is an optimal agent policy, as it gives the maximum worst-case expected state value of 100 in this case.

## C  Stage-wise Equilibrium, Robust Total Nash Equilibrium, and Robust Agent Policy

In Theorem 4.3, it has been proven that a state-robust totally optimal agent policy does not always exist for SAMGs. This section explores alternative solution concepts for the agent policy in SAMGs. We begin by demonstrating the existence of a unique robust state value function for each agent in C.1. Building on this property, we establish the existence of a stage-wise equilibrium for each state in C.2. However, we show in C.3 that the robust total Nash equilibrium may not always exist. As an alternative, we propose the concept of a robust agent policy and demonstrate its existence in C.4.

We first give a review of the Nash equilibrium used in the literature. The Nash equilibrium is a widely used solution concept in game theory, first proposed by Nash in Nash (1951) for general-sum finite one-shot games. It states that each player selects the best response strategy to the others' strategies and no player would want to deviate from the equilibrium, as doing so would result in a worse utility. This concept was later extended to infinite games by Debreu (Debreu, 1952), Glicksberg (Glicksberg, 1952), and Fan (Fan, 1952). Markov games, which involve a sequential decision process in a two-player zero-sum setting, were first defined by Shapley in Shapley (1953). Fink extended the Nash equilibrium concept to Markov games in Fink (1964) and proved that an equilibrium point exists in n-player general-sum discounted Markov games. The uncertainty in transition dynamics of a Markov game was considered in Nilim & El Ghaoui (2005); Iyengar (2005) using a robust optimization approach, with independent proofs for the existence of the equilibrium point. Additionally, uncertainty in utility (or "reward" in reinforcement learning) was also taken into account in Kardeş et al. (2011) for n-player finite state/action discounted Markov games, with a proof for the existence of the equilibrium point.

Despite the extensive study of the Nash equilibrium in game theory, the uncertainty in the state has not yet been explored in the context of Markov games. To the best of our knowledge, we are the first to formulate the problem of n-player finite state/action discounted Markov games with state uncertainty and to demonstrate the existence of a stage-wise equilibrium, as well as the non-existence of a robust total Nash equilibrium.

We use the following Assumption C.1 throughout this section.

**Assumption C.1.** The global state set $\mathcal{S}$ and the global action set $\mathcal{A}$ are finite sets.

### C.1 Unique Robust State Value Function

Denote the agent policies and adversary policies of all other agents and adversaries except agent $i$ and adversary $i$ as $\pi^{-i}$ and $\chi^{-i}$ respectively. We show that there exists a unique robust state value function for agent $i$ given any $\pi^{-i}$ and $\chi^{-i}$.

**Definition 4.4 (Robust state value function).** A state value function $V^i_{*,\pi^{-i},*,\chi^{-i}} : \mathcal{S} \to \mathbb{R}$ for agent $i$ given $\pi^{-i}$ and $\chi^{-i}$ is called a robust state value function if for all $s \in \mathcal{S}$,

$$V^i_{*,\pi^{-i},*,\chi^{-i}}(s) = \max_{\pi^i} \min_{\chi^i} \sum_{\rho \in \mathcal{P}_s} \chi(\rho|s) \sum_{a \in \mathcal{A}} \pi(a|\rho)$$

$$\left( r(s,a) + \gamma \sum_{s' \in \mathcal{S}} p(s'|s,a) V^i_{*,\pi^{-i},*,\chi^{-i}}(s') \right). \tag{21}$$

Note that we use $\pi(a|\rho) = \Pi^n_{i=1} \pi^i(a^i|\rho^i)$ to denote the joint agent policy. We use $\chi(\rho|s) = \Pi^n_{i=1} \chi^i(\rho^i|s)$ to denote the joint adversary policy.

Before proving the existence of the unique robust state value function, we first introduce some notations for this proof. For a given state value function $V^i_{*,\pi^{-i},*,\chi^{-i}} : \mathcal{S} \to \mathbb{R}$ defined on a finite state set $\mathcal{S}$, we can construct a state value vector $v^i = \mathrm{vec}(V^i_{*,\pi^{-i},*,\chi^{-i}}) = [V^i_{*,\pi^{-i},*,\chi^{-i}}(s)]_{s \in \mathcal{S}} \in \mathcal{V} := \mathbb{R}^{|\mathcal{S}|}$ by traversing all states, where $\mathrm{vec}(\cdot)$ is a vectorization function. The infinity norm on $\mathcal{V}$ is $\|v^i\|_\infty = \max_{s \in \mathcal{S}} |V^i(s)|$. Define the total expected return in state $s$ for $\pi^i$ and $\chi^i$ as

$$f^i_s(v^i, \pi^i, \pi^{-i}, \chi^i, \chi^{-i}) = \sum_{\rho \in \mathcal{P}_s} \chi(\rho|s) \sum_{a \in \mathcal{A}} \pi(a|\rho)$$

$$\left( r(s,a) + \gamma \sum_{s' \in \mathcal{S}} p(s'|s,a) [\mathrm{vec}^{-1}(v^i)](s') \right), \tag{22}$$

where $\pi^{-i}$ and $\chi^{-i}$ denotes the agent policies and the adversary policies of all other agents except agent $i$.

Define the robust state value in state $s$ given $\pi^{-i}$ and $\chi^{-i}$ as a function $\psi^i_s : \mathcal{V} \to \mathbb{R}$,

$$\psi^i_s(v^i, \pi^{-i}, \chi^{-i}) = \max_{\pi^i} \min_{\chi^i} f^i_s(v^i, \pi^i, \pi^{-i}, \chi^i, \chi^{-i}). \tag{23}$$

Note that $\psi^i_s$ gives a real number that denotes the total expected return in state $s$ given $\pi^{-i}$ and $\chi^{-i}$. We can construct a mapping $\Psi^i_{\pi,\chi} : \mathcal{V} \to \mathcal{V}$ from any state value vector $v^i$ to a robust state value vector $[\Psi^i_{\pi,\chi}(v^i)]_{s \in \mathcal{S}}$ by traversing all $s$, that is to say, $[\Psi^i_{\pi,\chi}(v^i)]_{s \in \mathcal{S}} = \psi^i_s(v^i, \pi^{-i}, \chi^{-i})$.

**Lemma C.2.** *For any $i \in \mathcal{N}$, the function $\Psi^i_{\pi,\chi} : \mathcal{V} \to \mathcal{V}$ is a contraction mapping given any $\pi^{-i}$ and $\chi^{-i}$ of other agents and adversaries except agent $i$ and adversary $i$.*

*Proof.* Let us consider two vectors $v^i, z^i \in \mathcal{V}$. For any $i \in \mathcal{N}$, given any $\pi^{-i}$ and $\chi^{-i}$, for all $s \in \mathcal{S}$, we have

$$\psi^i_s(v^i, \pi^{-i}, \chi^{-i}) = \max_{\pi^i} \min_{\chi^i} f^i_s(v^i, \pi^i, \pi^{-i}, \chi^i, \chi^{-i})$$

$$= f^i_s(v^i, \pi^{i*}, \pi^{-i}, \chi^{i*}, \chi^{-i}), \tag{24}$$

where $\pi^{i*}$ is the corresponding maximizer, and $\chi^{i*}$ is the corresponding optimizer for $\pi^{i*}$. Similarly, with the optimizers $\omega^{i*}$ and $\varphi^{i*}_1$ for the following maximin optimization problem, we have

$$\psi^i_s(z^i, \pi^{-i}, \chi^{-i}) = \max_{\omega^i} \min_{\varphi^i} f^i_s(z^i, \omega^i, \pi^{-i}, \varphi^i, \chi^{-i})$$

$$= f^i_s(z^i, \omega^{i*}, \pi^{-i}, \varphi^{i*}_1, \chi^{-i})$$

$$\geq f^i_s(z^i, \pi^{i*}, \pi^{-i}, \varphi^{i*}_2, \chi^{-i}), \tag{25}$$

where

$$\varphi_2^{i*} = \arg\min_{\varphi^i} f_s^i(z^i, \pi^{i*}, \pi^{-i}, \varphi^i, \chi^{-i}). \tag{26}$$

Then, for any $i \in \mathcal{N}$, given any $\pi^{-i}$ and $\chi^{-i}$, for all $s \in \mathcal{S}$, it holds that

$$
\begin{aligned}
&\psi_s^i(v^i, \pi^{-i}, \chi^{-i}) - \psi_s^i(z^i, \pi^{-i}, \chi^{-i}) \\
&= f_s^i(v^i, \pi^{i*}, \pi^{-i}, \chi^{i*}, \chi^{-i}) - f_s^i(z^i, \omega^{i*}, \pi^{-i}, \varphi_1^{i*}, \chi^{-i}) \\
&\le f_s^i(v^i, \pi^{i*}, \pi^{-i}, \chi^{i*}, \chi^{-i}) - f_s^i(z^i, \pi^{i*}, \pi^{-i}, \varphi_2^{i*}, \chi^{-i}) \\
&\le f_s^i(v^i, \pi^{i*}, \pi^{-i}, \varphi_2^{i*}, \chi^{-i}) - f_s^i(z^i, \pi^{i*}, \pi^{-i}, \varphi_2^{i*}, \chi^{-i}) \\
&= \sum_{\rho \in \mathcal{P}_s} \varphi_2^{i*}(\rho^i|s) \prod_{j\ne i} \chi^j(\rho^j|s) \sum_{a\in\mathcal{A}} \pi^{i*}(a^i|\rho^i) \times \\
&\quad \prod_{k\ne i} \pi^k(a^k|\rho^k) \left( r + \gamma \sum_{s'\in\mathcal{S}} p(s'|s, a)[\mathrm{vec}^{-1}(v^i)](s') \right) \\
&\quad - \sum_{\rho\in\mathcal{P}_s} \varphi_2^{i*}(\rho^i|s) \prod_{j\ne i} \chi^j(\rho^j|s) \sum_{a\in\mathcal{A}} \pi^{i*}(a^i|\rho^i) \times \\
&\quad \prod_{k\ne i} \pi^k(a^k|\rho^k) \left( r + \gamma \sum_{s'\in\mathcal{S}} p(s'|s, a)[\mathrm{vec}^{-1}(z^i)](s') \right) \\
&= \sum_{\rho\in\mathcal{P}_s} \varphi_2^{i*}(\rho^i|s) \prod_{j\ne i} \chi^j(\rho^j|s) \sum_{a\in\mathcal{A}} \pi^{i*}(a^i|\rho^i) \times \\
&\quad \prod_{k\ne i} \pi^k(a^k|\rho^k)\gamma \sum_{s'\in\mathcal{S}} p(s'|s, a) \times \\
&\quad \left\{ [\mathrm{vec}^{-1}(v^i)](s') - [\mathrm{vec}^{-1}(z^i)](s') \right\} \\
&\le \sum_{\rho\in\mathcal{P}_s} \varphi_2^{i*}(\rho^i|s) \prod_{j\ne i} \chi^j(\rho^j|s) \sum_{a\in\mathcal{A}} \pi^{i*}(a^i|\rho^i) \times \\
&\quad \prod_{k\ne i} \pi^k(a^k|\rho^k)\gamma \sum_{s'\in\mathcal{S}} p(s'|s, a)\|v^i - z^i\|_\infty \\
&= \gamma\|v^i - z^i\|_\infty. \tag{27}
\end{aligned}
$$

The second inequality in Eq. (27) follows

$$\chi^{i*} = \arg\min_{\chi^i} f_s^i(v^i, \pi^{i*}, \pi^{-i}, \chi^i, \chi^{-i}). \tag{28}$$

Because for any $i \in \mathcal{N}$, given any $\pi^{-i}$ and $\chi^{-i}$, for all $s \in \mathcal{S}$

$$\psi_s^i(v^i, \pi^{-i}, \chi^{-i}) - \psi_s^i(z^i, \pi^{-i}, \chi^{-i}) \le \gamma\|v^i - z^i\|_\infty, \tag{29}$$

Based on symmetry, we have

$$
\begin{aligned}
\psi_s^i(z^i, \pi^{-i}, \chi^{-i}) - \psi_s^i(v^i, \pi^{-i}, \chi^{-i}) &\le \gamma\|z^i - v^i\|_\infty \\
&= \gamma\|v^i - z^i\|_\infty. \tag{30}
\end{aligned}
$$

Thus, it holds that for any $i \in \mathcal{N}$, given any $\pi^{-i}$ and $\chi^{-i}$

$$\|\Psi_{\pi,\chi}^i(v^i) - \Psi_{\pi,\chi}^i(z^i)\|_\infty \le \gamma\|v^i - z^i\|_\infty, \tag{31}$$

that is to say, the function $\Psi_{\pi,\chi}^i$ is a contraction mapping. □

**Theorem 4.5 (Existence of Unique Robust State Value Function).** *For an SAMG with finite state and finite action spaces, for any $i \in \mathcal{N}$, given any $\pi^{-i}$ and $\chi^{-i}$ of other agents and adversaries except agent $i$ and adversary $i$, there exists a unique robust state value function $V^i_{*,\pi^{-i},*,\chi^{-i}} : \mathcal{S} \to \mathbb{R}$ for agent $i$ such that for all $s \in \mathcal{S}$,*

$$V^i_{*,\pi^{-i},*,\chi^{-i}}(s) = \max_{\pi^i} \min_{\chi^i} \sum_{\rho \in \mathcal{P}_s} \chi(\rho|s) \sum_{a \in \mathcal{A}} \pi(a|\rho)$$

$$\left( r(s,a) + \gamma \sum_{s' \in \mathcal{S}} p(s'|s,a) V^i_{*,\pi^{-i},*,\chi^{-i}}(s') \right). \tag{32}$$

*Proof.* For any $i \in \mathcal{N}$, there exists a state value function $V^i_{*,\pi^{-i},*,\chi^{-i}}$ satisfying (32) if and only if $v^i = \text{vec}(V^i_{*,\pi^{-i},*,\chi^{-i}})$ is a fixed point of $\Psi^i_{\pi,\chi} : \mathcal{V} \to \mathcal{V}$, where $[\Psi^i_{\pi,\chi}(v^i)]_{s \in \mathcal{S}} = \psi^i_s(v^i, \pi^{-i}, \chi^{-i})$ and $\psi^i_s(v^i, \pi^{-i}, \chi^{-i})$ is defined in (23). We use Banach's fixed point theorem to prove this as follows.

Because any finite-dimensional normed vector space is complete (Kreyszig, 1991), the $(\mathcal{V}, \|\cdot\|_\infty)$ is a complete Banach space. Also, for any $i \in \mathcal{N}$, given any $\pi^{-i}$ and $\chi^{-i}$, the function $\Psi^i_{\pi,\chi}$ is a contraction mapping according to Lemma C.2. Therefore, by Banach's fixed point theorem, there is a unique fixed point $v^i$ such that $\Psi^i_{\pi,\chi}(v^i) = v^i$. In other words, for any $i \in \mathcal{N}$, given any $\pi^{-i}$ and $\chi^{-i}$, there exists a unique $V^i_{*,\pi^{-i},*,\chi^{-i}}$ such that

$$V^i_{*,\pi^{-i},*,\chi^{-i}}(s) = \max_{\pi^i} \min_{\chi^i} f^i_s(v^i, \pi^i, \pi^{-i}, \chi^i, \chi^{-i}). \tag{33}$$

$\square$

Denote the state value function for agent $i$ given any $\pi^{-i}$ and $\chi^{-i}$ of other agents and adversaries except agent $i$ and adversary $i$ as

$$V^i_{\pi^i,\pi^{-i},\chi^i,\chi^{-i}}(s) = f^i_s(v^i, \pi^i, \pi^{-i}, \chi^i, \chi^{-i}), \tag{34}$$

where $v^i = \text{vec}(V^i_{*,\pi^{-i},*,\chi^{-i}})$. Then we have the following corollary for Theorem C.1.

**Corollary C.3.** *For an SAMG with finite state and finite action spaces, let $V^i_{*,\pi^{-i},*,\chi^{-i}}$ be the unique robust state value function for agent $i$ given any $\pi^{-i}$ and $\chi^{-i}$ such that for all $s \in \mathcal{S}$,*

$$V^i_{*,\pi^{-i},*,\chi^{-i}}(s) = \max_{\pi^i} \min_{\chi^i} f^i_s(v^i, \pi^i, \pi^{-i}, \chi^i, \chi^{-i})$$

$$= f^i_s(v^i, \pi^{i*}, \pi^{-i}, \chi^{i*}, \chi^{-i}), \tag{35}$$

*where $v^i = \text{vec}(V^i_{*,\pi^{-i},*,\chi^{-i}})$, $\pi^{i*}$ is the corresponding maximizer at state $s$, and $\chi^{i*}$ is the corresponding optimizer for $\pi^{i*}$ at state $s$, then for state $s$ it holds that $V^i_{\pi^{i*},\pi^{-i},\chi^{i*},\chi^{-i}}(s) \geq V^i_{\pi^i,\pi^{-i},\chi^{i*},\chi^{-i}}(s)$ for any $\pi^i$, and $V^i_{\pi^{i*},\pi^{-i},\chi^{i*},\chi^{-i}}(s) \leq V^i_{\pi^{i*},\pi^{-i},\chi^i,\chi^{-i}}(s)$ for any $\chi^i$.*

## C.2 Existence of the Stage-wise Equilibrium

Before we show the existence of the robust total Nash equilibrium, we first show a concept of the stage-wise equilibrium.

**Definition C.4 (Stage-wise Equilibrium).** For an SAMG, the policy $(\pi^*, \chi^*)$ is a stage-wise equilibrium for state $s$ if for all $i \in \mathcal{N}$ and all $\pi^i$ and $\chi^i$, it holds that

$$V^i_{\pi^i,\pi^{-i*},\chi^{i*},\chi^{-i*}}(s) \leq V^i_{\pi^{i*},\pi^{-i*},\chi^{i*},\chi^{-i*}}(s)$$

$$\leq V^i_{\pi^{i*},\pi^{-i*},\chi^i,\chi^{-i*}}(s), \tag{36}$$

where $\pi^{-i}$ and $\chi^{-i}$ denotes the agent policies and adversary policies of all the other agents except agent $i$, respectively.

The Nash equilibrium was originally proposed by Nash for finite one-shot games, in which the state transition of the environment is not considered. When the concept of Nash equilibrium is extended to Markov games, the existence of the equilibrium is shown through the existence of a state-wise equilibrium for each state. A policy that is a stage-wise equilibrium for all states is considered a Nash equilibrium for the Markov game.

This idea brings the following proposition to show the relationship between the robust total Nash equilibrium and the stage-wise equilibrium for SAMGs.

**Proposition C.5.** *The policy* $(\pi^*, \chi^*)$ *is a robust total Nash equilibrium for an SAMG if the policy* $(\pi^*, \chi^*)$ *is a stage-wise equilibrium for all* $s \in \mathcal{S}$.

*Proof.* It is a natural result according to the Definition 4.6 and the Definition C.4. □

We show the existence of the stage-wise equilibrium defined in Definition C.4 in the following theorem.

**Theorem C.6** (**Existence of Stage-wise equilibrium**). *For SAMGs with finite state and finite action spaces, the stage-wise equilibrium defined in Definition C.4 exists for any* $s \in \mathcal{S}$.

*Proof.* Let us construct a $2n$ player game for any $s \in \mathcal{S}$. We have $n$ agents and $n$ adversaries in the player set. We introduce uniform notations for the agents and adversaries to describe a $2n$ player game at state $s$. The player set $\mathcal{I} = \{1, ..., n, n+1, ..., 2n\}$. The first half of the player set $\{1, ..., n\}$ represents agents, while the second half $\{n+1, ..., 2n\}$ represents adversaries. The set of available actions for player $i$ is

$$A_s^i = \begin{cases} \underbrace{\mathcal{A}^i \times \mathcal{A}^i \cdots \times \mathcal{A}^i}_{\text{total number: } |\mathcal{P}_s^i|}, & i = 1, ..., n; \\ \mathcal{P}_s^{i-n}, & i = n+1, ..., 2n. \end{cases} \tag{37}$$

Each adversary's action set includes all possible perturbed states in the admissible perturbed state set at state $s$. Each agent's action set includes all possible joint actions given every possible perturbed state. Take the two-agent two-state game in Fig. 8 as an example, the player set $\mathcal{I} = \{1, 2, 3, 4\}$. Player 3 is the adversary for agent player 1. Player 4 is the adversary for agent player 2. If the current true state is $s_1$, then $A_{s_1}^1 = A_{s_1}^2 = \{(a_1, a_1), (a_1, a_2), (a_2, a_2), (a_2, a_1)\}$ are the action sets for two agent players. In $A_{s_1}^1$ for agent 1, the joint action $(a_1, a_2)$ means selecting $a_1$ if the perturbed state for agent 1 is $s_1$ and selecting $a_2$ if the perturbed state for agent 1 is $s_2$. For two adversary players, $A_{s_1}^3 = A_{s_1}^4 = \{s_1, s_2\}$, as adversaries can perturb the true state $s_1$ to $s_2$.

We consider the mixed strategy $\sigma_s^i \in \Delta(A_s^i)$ for player $i$. Note that the mixed strategy for each adversary gives us the probability distribution of all possible perturbed states for state $s$, i.e. $\chi^{i-n}(\rho^{i-n}|s) = \sigma_s^i(\rho^{i-n})$ for $i = n+1, ..., 2n$. Then we show how we can get each agent's policy $\pi^i(a^i|\rho^i)$ based on its mixed strategy $\sigma_s^i$ by calculating the marginal probabilities. Denote the total number of possible perturbed state for agent $i$ at state $s$ as $P$ such that $P = |\mathcal{P}_s^i|$. Here we drop the subscript $s$ in $P_s$ for a concise representation. The perturbed state set for agent $i$ is represented as $\{\rho_1^i, \rho_2^i, \ldots, \rho_P^i\}$. Denote the joint action of agent $i$ as $b^i = (b_1^i, b_2^i, \ldots, b_P^i)$ where $b_k^i$ is the action selected for the perturbed state $\rho_k^i \in \mathcal{P}_s^i$. Then the mixed strategy $\sigma_s^i(b_1^i, b_2^i, ..., b_P^i)$ gives us the joint probability of selecting $b_k^i$ for $\rho_k^i$ for all $k = 1, 2, ..., P$. We can get the marginal probability of selecting action $a^i$ given the perturbed state $\rho_k^i \in \mathcal{P}_s^i$ as

$$\pi^i(a^i|\rho_k^i) = \sum_{\{b^i \in A_s^i | b_k^i = a^i\}} \sigma_s^i(b_1^i, b_2^i, ..., b_P^i). \tag{38}$$

The marginal probability of selecting action $a^i$ given the perturbed state $\rho_k^i$ is calculated by summing up the joint probability over all joint actions in which agent $i$ selects $a^i$ given the perturbed state $\rho_k^i$. Take the two-agent two-state game in Fig. 8 as an example, if the current perturbed state for agent 1 is $\rho^1 = s_1$, then agent 1's policy is

$$\pi^1(a_1|\rho^1 = s_1) = \sigma^1(a_1, a_1) + \sigma^1(a_1, a_2)$$
$$\pi^1(a_2|\rho^1 = s_1) = \sigma^1(a_2, a_1) + \sigma^1(a_2, a_2). \tag{39}$$

Note that the mixed strategy $\sigma_s^i \in \Delta(A_s^i)$ only gives part of the agent and adversary policies. For example, the mixed strategy for the adversaries only gives a distribution of the perturbed states for $s_t = s$. We construct the complete agent and adversary policies as follows: For $i = 1, ..., n$, the agent $i$'s policy is

$$\pi^i(a^i|\rho^i) = \begin{cases} \sum_{\{b^i \in A_s^i | b_k^i = a^i\}} \sigma_s^i(b_1^i, b_2^i, ..., b_P^i), \\ \text{for } \rho^i = \rho_k^i \in \mathcal{P}_s^i; \\ \mathcal{U}(\mathcal{A}^i), \\ \text{for } \rho^i \notin \mathcal{P}_s^i, \end{cases} \tag{40}$$

where $\mathcal{U}(\mathcal{A}^i)$ represents a uniform distribution on $\mathcal{A}^i$. For $i = 1, ..., n$, the adversary $i$'s policy is

$$\chi^i(\rho^i|s_t) = \begin{cases} \sigma_s^{i+n}(\rho^i), & \text{for } s_t = s; \\ \mathcal{U}(\mathcal{P}_s^i), & \text{for } s_t \neq s, \end{cases} \tag{41}$$

where $\mathcal{U}(\mathcal{P}_s^i)$ represents a uniform distribution on $\mathcal{P}_s^i$.

The utility function for player $i$ is

$$u_s^i(\sigma_s^i, \sigma_s^{-i}) = \begin{cases} f_s^i(v^{i*}, \pi^i, \pi^{-i}, \chi^i, \chi^{-i}), \\ \text{for } i = 1, ..., n; \\ -f_s^{i-n}(v^{(i-n)*}, \pi^{i-n}, \pi^{-(i-n)}, \\ \chi^{i-n}, \chi^{-(i-n)}), \\ \text{for } i = n+1, ..., 2n. \end{cases} \tag{42}$$

where $\sigma_s^{-i}$ denotes the strategies of all other players except player $i$, $v^{i*} = \text{vec}(V_{*,\pi^{-i},*,\chi^{-i}}^i)$, and $V_{*,\pi^{-i},*,\chi^{-i}}^i$ is the unique robust state value function of agent $i$ when the policies of other agents and adversaries are given by $\pi^{-i}$ and $\chi^{-i}$. The $v^{i*}$ satisfies

$$[\text{vec}^{-1}(v^{i*})](s) = \max_{\pi^i} \min_{\chi^i} f_s^i(v^{i*}, \pi^i, \pi^{-i}, \chi^i, \chi^{-i}), \tag{43}$$

where $f_s^i$ is defined for player $i$ in (22) as

$$f_s^i(v^i, \pi^i, \pi^{-i}, \chi^i, \chi^{-i}) = \sum_{\rho \in \mathcal{P}_s} \chi(\rho|s) \sum_{a \in \mathcal{A}} \pi(a|\rho)$$
$$\left( r(s, a) + \gamma \sum_{s' \in \mathcal{S}} p(s'|s, a)[\text{vec}^{-1}(v^i)](s') \right).$$

Note that $\sigma_s^{-i}$ includes both $\pi^{-i}$ and $\chi^{-i}$ for any $i \in \mathcal{I}$, and the existence of $V_{*,\pi^{-i},*,\chi^{-i}}^i$ is guaranteed by Theorem C.1. Thus, the utility function is well-defined.

Since the state set $\mathcal{S}$ is finite, $\mathcal{P}_s^i \subseteq \mathcal{S}$ is a finite set for all $i \in \mathcal{N}$. Also, $\mathcal{A}^i$ is a finite set for all $i \in \mathcal{N}$. Therefore, $\Delta(A_s^i)$ is compact and convex for all $i \in \mathcal{I}$. Moreover, for all $i \in \mathcal{I}$, $u_s^i(\sigma_s^i, \cdot)$ is linear in $\sigma_s^i$ and therefore continuous and concave in $\sigma_s^i$. According to the theorem (Debreu Debreu (1952), Glicksberg Glicksberg (1952), Fan Fan (1952)), the conditions for the existence of a Nash Equilibrium are satisfied, hence, there exists a Nash equilibrium $\sigma_s^*$ for this $2n$ player game for any $s \in \mathcal{S}$ such that for any $i \in \mathcal{I}$, $u_s^i(\sigma_s^{i*}, \sigma_s^{-i*}) \geq u_s^i(\sigma_s^i, \sigma_s^{-i*})$ for any $\sigma_s^i$.

Denote the agent and adversary policies as $(\pi^*, \chi^*)$ that are constructed following Eq. (40) and Eq. (41) by plugging in the Nash equilibrium $(\sigma_s^{i*}, \sigma_s^{-i*})$. Substituting the $(\pi^*, \chi^*)$ into $u_s^i(\sigma_s^{i*}, \sigma_s^{-i*}) \geq u_s^i(\sigma_s^i, \sigma_s^{-i*})$ and plugging in the definition of the utility functions, for any $i = 1, 2, ..., n$, it holds that

$$f_s^i(v^{i*}, \pi^{i*}, \pi^{-i*}, \chi^{i*}, \chi^{-i*}) \geq f_s^i(v^{i*}, \pi^i, \pi^{-i*}, \chi^{i*}, \chi^{-i*}), \tag{44}$$

for any $\pi^i$. Also, for any $i = 1, 2, ..., n$, it holds that

$$f_s^i(v^{i*}, \pi^{i*}, \pi^{-i*}, \chi^{i*}, \chi^{-i*}) \leq f_s^i(v^{i*}, \pi^{i*}, \pi^{-i*}, \chi^i, \chi^{-i*}), \tag{45}$$

for any $\chi^i$. Therefore,

$$\max_{\pi^i} \min_{\chi^i} f_s^i(v^{i*}, \pi^i, \pi^{-i*}, \chi^i, \chi^{-i*})$$
$$= f_s^i(v^{i*}, \pi^{i*}, \pi^{-i*}, \chi^{i*}, \chi^{-i*}). \tag{46}$$

According to Corollary C.3, for any $\pi^i$, it holds that

$$V_{\pi^{i*}, \pi^{-i*}, \chi^{i*}, \chi^{-i*}}^i(s) \geq V_{\pi^i, \pi^{-i*}, \chi^{i*}, \chi^{-i*}}^i(s), \tag{47}$$

Also, for any $\chi^i$, it holds that

$$V_{\pi^{i*}, \pi^{-i*}, \chi^{i*}, \chi^{-i*}}^i(s) \leq V_{\pi^{i*}, \pi^{-i*}, \chi^i, \chi^{-i*}}^i(s). \tag{48}$$

Thus, the stage-wise equilibrium defined in Definition C.4 exists for any $s \in \mathcal{S}$. $\qquad\square$

### C.3 Non-existence of Robust Total Nash Equilibrium

Theorem C.6 demonstrates the existence of a stage-wise equilibrium for any state $s \in \mathcal{S}$. In classic Markov games (Fink, 1964) and Markov games with reward/transition uncertainties (Kardeş et al., 2011; Nilim & El Ghaoui, 2005; Iyengar, 2005), this result naturally extends to the existence of a Nash equilibrium policy, as all agents' and adversaries' policies are based on the current true state. If a stage-wise equilibrium exists for any state $s \in \mathcal{S}$, then a Nash equilibrium can be constructed by taking the policies for each state $s$ from their corresponding stage-wise equilibrium for state $s$ (Fink, 1964; Kardeş et al., 2011; Nilim & El Ghaoui, 2005; Iyengar, 2005). However, this natural extension cannot be used for our SAMG problem because the agent's policy is based on the perturbed state instead of the true state. The problem is that the agent's stage-wise equilibrium in one state may not be consistent with its stage-wise equilibrium in a different state. We illustrate this idea in the following theorem to show that the robust total Nash equilibrium does not always exist for SAMGs.

**Theorem 4.7 (Non-existence of Robust Total Nash Equilibrium).** *For SAMGs with finite state and finite action spaces, the robust total Nash equilibrium defined in Definition 4.6 does not always exist.*

*Proof.* We prove this theorem by showing that the following two-agent two-state game in Fig. 9 does not have a robust total Nash equilibrium. The two-agent two-state game in Fig. 9 is basically the same as the

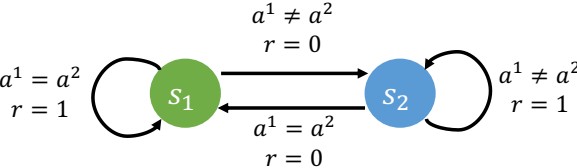

Figure 9: A new two-agent two-state game example. Agents get reward 1 at state $s_1$ if they choose the same action. Agents get reward 1 at state $s_2$ if they choose different actions.

two-agent two-state game in Fig. 8. The only difference is we changed the state transition for the state $s_1$. The new state transition functions for the state $s_1$ are

$$p(s' = s_2 | s = s_1, a^1 \neq a^2) = 1,$$
$$p(s' = s_1 | s = s_1, a^1 = a^2) = 1. \tag{49}$$

We first consider the stage-wise equilibriums for each state.

For state $s_1$, the stage-wise equilibrium requires $\Pr(a_t^1 = a_t^2) = 1$ for all $t$. One example of the agent policy is $\pi^1(a_1|s_1) = \pi^1(a_1|s_2) = \pi^2(a_1|s_1) = \pi^2(a_1|s_2) = 1$. Note that the agent should have a policy for both $s_1$ and $s_2$ even when considering the state-wise equilibrium for the state $s_1$ (This means the current true state is $s_1$).

This is because the adversary can perturb each agent's state observation to be $s_2$. There is no requirement for the adversary policy in the state-wise equilibrium because when $\Pr(a_t^1 = a_t^2) = 1$, the true state never transits. The state value for $s_1$ is $V(s_1) = 100$.

Similarly, for state $s_2$, the stage-wise equilibrium requires $\Pr(a_t^1 \neq a_t^2) = 1$ for all $t$. One example of the agent policy is $\pi^1(a_1|s_1) = \pi^1(a_1|s_2) = \pi^2(a_2|s_1) = \pi^2(a_2|s_2) = 1$. There is no requirement for the adversary policy in the state-wise equilibrium of $s_2$. The state value for $s_2$ is $V(s_2) = 100$.

Since the stage-wise equilibriums have conflict requirements for the agent policy in $s_1$ and $s_2$, there is no agent policy satisfying the requirements of the stage-wise equilibriums in both $s_1$ and $s_2$ at the same time. Therefore, there is no robust total Nash equilibrium for agents in this two-agent two-state game. □

In the proof of Theorem 4.7, we intended to present a straightforward example for ease of understanding. However, more counter-examples can be more illustrative in demonstrating the prevalence of non-existence scenarios. As long as the two stage-wise equilibriums have different requirements (not necessarily contrary to each other), there is no Nash equilibrium.

To elaborate, consider a 2-state 2-action game. If, for state $s_1$, the stage-wise equilibrium necessitates choosing action $a^1$ with probability 0.2 and $a^2$ with probability 0.8, and for state $s_2$, the stage-wise equilibrium requires choosing $a^1$ with probability 0.6 and $a^2$ with probability 0.4, then it's clear that no Nash equilibrium can simultaneously satisfy these requirements. This example illustrates the absence of a robust Nash equilibrium in such a 2-state 2-action game scenario.

Our conclusion is similar to that of Theorem 4.3, in that it is not always possible to find a policy that is a stage-wise equilibrium for all states. When facing adversarial state perturbations, trade-offs must be made among different states. As a result, the traditional solution concepts of an optimal agent policy and the robust total Nash equilibrium cannot be applied to SAMGs.

## C.4 Existence of Robust Agent Policy

We need to consider a new objective that is not state-dependent. Therefore, we propose a new objective, the worst-case expected state value, in Definition 4.8 as

$$\mathbb{E}_{s_0 \sim \Pr(s_0)}\left[\bar{V}_\pi(s_0)\right],$$

where $\Pr(s_0)$ is the probability distribution of the initial state.

The new objective of "worst-case expected state value" is designed specifically for the state perturbation problem present in SAMGs. It is proposed as a response to our analysis of the non-existence of widely-used concepts. We demonstrate that these concepts can be easily corrupted by adversaries, requiring agents to make trade-offs between different states. This is the reason for introducing the new objective. The agent policy that aims to maximize this worst-case expected state value is referred to as a robust agent policy.

In this section, we show the existence of a robust agent policy to maximize the worst-case expected state value. We first introduce lemmas for this proof.

Denote $p^{\pi,\chi,s_0}(s_t)$ as the probability of reaching state $s_t$ given the agent policy $\pi$, adversary policy $\chi$, and initial state $s_0$. Let $p^{\pi,\chi,s_0}(s_0) = 1$. The connection between $p^{\pi,\chi,s_0}(s_{t+1})$ and $p^{\pi,\chi,s_0}(s_t)$ is:

$$p^{\pi,\chi,s_0}(s_{t+1}) =$$
$$\sum_{s_t \in \mathcal{S}} \sum_{a_t \in \mathcal{A}} \sum_{\rho_t \in \mathcal{P}} p(s_{t+1}|s_t, a_t)\pi(a_t|\rho_t)\chi(\rho_t|s_t)p^{\pi,\chi,s_0}(s_t). \tag{50}$$

For a concise representation, we omit the subscript $s_t$ of $\mathcal{P}_{s_t}$ in this section. Consider the function

$$g_t^{s_0}(\pi, \chi) = \sum_{s_t \in \mathcal{S}} \sum_{a_t \in \mathcal{A}} \sum_{\rho_t \in \mathcal{P}} p^{\pi,\chi,s_0}(s_t) \times$$
$$\pi(a_t|\rho_t)\chi(\rho_t|s_t)\gamma^t r_{t+1}(s_t, a_t). \tag{51}$$

**Lemma C.7.** *The function $g_t^{s_0}$ is continuous on $\Delta(\mathcal{A}) \times \Delta(\mathcal{P})$ for any $t = 0, 1, 2, ..., n$ where $n \in \mathbb{N}_+$.*

*Proof.* To prove the continuity, we construct some equivalent vectors as follows. We define a vector $\vec{\pi} \in \mathbb{R}^{|\mathcal{A}||\mathcal{P}|}$ and $\vec{\pi}(a, \rho) = \pi(a|\rho)$ for $a \in \mathcal{A}, \rho \in \mathcal{P}$, and a vector $\vec{\chi} \in \mathbb{R}^{|\mathcal{P}||\mathcal{S}|}$ where $\vec{\chi}(\rho, s) = \chi(\rho|s)$ for $\rho \in \mathcal{P}, s \in \mathcal{S}$. And a vector constant $\vec{r} \in \mathbb{R}^{|\mathcal{S}||\mathcal{A}|}$ where $\vec{r}(s, a) = r(s, a)$.

$$
\begin{aligned}
\vec{\pi}^\top &= [\pi(a^1|\rho^1), \cdots, \pi(a^{|\mathcal{A}|}|\rho^1), \pi(a^2|\rho^1), \cdots, \pi(a^{|\mathcal{A}|}|\rho^{|\mathcal{P}|})] \\
\vec{\chi}^\top &= [\chi(\rho^1|s^1), \cdots, \chi(\rho^{|\mathcal{P}|}|s^1), \chi(\rho^2|s^{|\mathcal{S}|}), \cdots, \chi(\rho^{|\mathcal{P}|}|s^{|\mathcal{S}|})] \\
\vec{p}_t &= [p^{\pi,\chi,s_0}(s_t = s^1), \cdots, p^{\pi,\chi,s_0}(s_t = s^{|\mathcal{S}|})]
\end{aligned}
\tag{52}
$$

Note that when $\rho \notin \mathcal{P}_s$, then the entry $\chi(\rho|s) = 0$. $\vec{p}_t \in \mathbb{R}^{|S|}$ can be expressed as a linear combination of $\vec{p}_{t-1}, \vec{\pi}$ and $\vec{\chi}$ according to (50). Let's first consider the case $t = 0$,

$$
g_0^{s_0}(\pi, \chi) = \sum_{a_0 \in \mathcal{A}} \sum_{\rho_0 \in \mathcal{P}} \pi(a_0|\rho_0)\chi(\rho_0|s_0)r(s_0, a_0)
\tag{53}
$$

Function $g_0^{s_0}$ can be expressed as a linear combination of $\vec{r}, \vec{\pi}$ and $\vec{\chi}$. We consider the general case

$$
\begin{aligned}
g_t^{s_0}(\pi, \chi) = \sum_{s_t \in \mathcal{S}} \sum_{a_t \in \mathcal{A}} \sum_{\rho_t \in \mathcal{P}} p^{\pi,\chi,s_0}(s_t) \times \\
\pi(a_t|\rho_t)\chi(\rho_t|s_t)\gamma^t r_{t+1}(s_t, a_t).
\end{aligned}
\tag{54}
$$

Function $g_t^{s_0}$ can be expressed as a linear combination of $\vec{r}, \vec{p}_t, \vec{\pi}$ and $\vec{\chi}$. Therefore, $g_t^{s_0}$ is continuous on $\Delta(\mathcal{A}) \times \Delta(\mathcal{P})$ for any $t = 0, 1, 2, ..., n$ where $n \in \mathbb{N}_+$. $\qquad\square$

**Lemma C.8.** *For any $s_0 \in \mathcal{S}$, the series $\{\sum_{t=0}^n g_t^{s_0}(\pi, \chi)\}, n = 1, 2, ...,$ converges uniformly on $\Delta(\mathcal{A}) \times \Delta(\mathcal{P})$.*

*Proof.* Consider $M_t^{s_0}(\pi, \chi) = \gamma^t R^{max}$, where $R^{max}$ is the largest absolute value of the rewards. We can check that $|g_t^{s_0}(\pi, \chi)| \le M_t^{s_0}(\pi, \chi)$ for $t \ge 0$ as follows.

$$
\begin{aligned}
&|g_t^{s_0}(\pi, \chi)| \\
&= \left| \sum_{s_t \in \mathcal{S}} \sum_{a_t \in \mathcal{A}} \sum_{\rho_t \in \mathcal{P}} p^{\pi,\chi,s_0}(s_t)\pi(a_t|\rho_t)\chi(\rho_t|s_t)\gamma^t r_{t+1}(s_t, a_t) \right| \\
&\le \left| \sum_{s_t \in \mathcal{S}} \sum_{a_t \in \mathcal{A}} \sum_{\rho_t \in \mathcal{P}} p^{\pi,\chi,s_0}(s_t)\pi(a_t|\rho_t)\chi(\rho_t|s_t)\gamma^t R^{max} \right| \\
&= \gamma^t R^{max} \times \left| \sum_{s_t \in \mathcal{S}} \sum_{a_t \in \mathcal{A}} \sum_{\rho_t \in \mathcal{P}} p^{\pi,\chi,s_0}(s_t)\pi(a_t|\rho_t)\chi(\rho_t|s_t) \right| \\
&= \gamma^t R^{max} \times \left| \sum_{s_t \in \mathcal{S}} \sum_{a_t \in \mathcal{A}} \sum_{\rho_t \in \mathcal{P}} \Pr(s_t, a_t, \rho_t \mid s_0, \pi, \chi) \right| \\
&= \gamma^t R^{max} \times 1 = M_t^{s_0}(\pi, \chi).
\end{aligned}
\tag{55}
$$

Meanwhile,

$$
\sum_{t=0}^{\infty} M_t^{s_0}(\pi, \chi) = \sum_{t=0}^{\infty} \gamma^t R^{max} = \frac{R^{max}}{1 - \gamma},
\tag{56}
$$

so $\sum g_t^{s_0}$ converges uniformly on $\Delta(\mathcal{A}) \times \Delta(\mathcal{P})$ according to the Weierstrass M-test in Theorem 7.10 of Rudin et al. (1976). $\qquad\square$

Lemma C.8 shows the series $\{\sum_{t=0}^n g_t^{s_0}(\pi, \chi)\}, n = 1, 2, ...,$ converges uniformly on $\Delta(\mathcal{A}) \times \Delta(\mathcal{P})$ for any $s_0 \in \mathcal{S}$. In the following lemma, we show $\sum_{t=0}^\infty g_t^{s_0}(\pi, \chi)$ is continuous on $\Delta(\mathcal{A}) \times \Delta(\mathcal{P})$ for any $s_0 \in \mathcal{S}$. Denote $h^{s_0}(\pi, \chi) = \sum_{t=0}^\infty g_t^{s_0}(\pi, \chi)$.

**Lemma C.9.** *The function $h^{s_0}$ is continuous on $\Delta(\mathcal{A}) \times \Delta(\mathcal{P})$ for any $s_0 \in \mathcal{S}$.*

*Proof.* Consider $h_n^{s_0}(\pi, \chi) = \sum_{t=0}^n g_t^{s_0}(\pi, \chi)$ for $n \in \mathbb{N}_+$. Since $h_n^{s_0}$ is a linear combination of $\{g_t^{s_0}\}_{t=0,1,2,\cdots,n}$ and $g_t^{s_0}$ is continuous on $\Delta(\mathcal{A}) \times \Delta(\mathcal{P})$ for any $t = 0, 1, 2, \cdots, n$ according to Lemma C.7, the sequence $\{h_n^{s_0}\}$ is a sequence of continuous functions on $\Delta(\mathcal{A}) \times \Delta(\mathcal{P})$. Meanwhile, $h_n^{s_0} \to h^{s_0}$ uniformly on $\Delta(\mathcal{A}) \times \Delta(\mathcal{P})$ for any $s_0 \in \mathcal{S}$ according to Lemma C.8, therefore $h^{s_0}$ is continuous on $\Delta(\mathcal{A}) \times \Delta(\mathcal{P})$ for any $s_0 \in \mathcal{S}$ according to the uniform limit theorem in Theorem 7.12 of Rudin et al. (1976). $\qquad\square$

The following theorem shows finding a robust agent policy is equivalent to solving a maximin problem.

**Theorem 4.10.** *Finding an agent policy $\pi$ to maximize the worst-case expected state value under an optimal adversary for $\pi$ is equivalent to the maximin problem: $\max_\pi \min_\chi \sum_{s_0} \Pr(s_0) V_{\pi,\chi}(s_0)$.*

*Proof.* According to the Proposition 4.1, for any fixed agent policy $\pi$, there exists an optimal adversary policy $\chi^*$ such that $\bar{V}_\pi(s_0) = \min_\chi V_{\pi,\chi}(s_0)$ for any $s_0 \in \mathcal{S}$. Thus,

$$
\max_\pi \mathbb{E}_{s_0 \sim \Pr(s_0)} \left[ \bar{V}_\pi(s_0) \right]
$$

$$
= \max_\pi \mathbb{E}_{s_0 \sim \Pr(s_0)} \left[ \min_\chi V_{\pi,\chi}(s_0) \right] \qquad \text{(Eq. (2))}
$$

$$
= \max_\pi \sum_{s_0} \Pr(s_0) \min_\chi V_{\pi,\chi}(s_0) \quad \text{(Definition of Expectation)}
$$

$$
= \max_\pi \min_\chi \sum_{s_0} \Pr(s_0) V_{\pi,\chi}(s_0), \qquad \text{(Proposition 4.1)} \tag{57}
$$

$\qquad\square$

Finally, we show the existence of the robust agent policy to maximize the worst-case expected state value in the following theorem.

**Theorem 4.11 (Existence of Robust Agent Policy).** *For SAMGs with finite state and finite action spaces, there exists a robust agent policy to maximize the worst-case expected state value defined in Definition 4.8.*

*Proof.* According to Theorem 4.10, finding an agent policy $\pi$ to maximize the worst-case expected state value under an optimal adversary for $\pi$ is equivalent to the following maximin problem:

$$
\max_\pi F(\pi)
$$

$$
:= \max_\pi \mathbb{E}_{s_0 \sim \Pr(s_0)} \left[ \bar{V}_\pi(s_0) \right]
$$

$$
= \max_\pi \min_\chi \sum_{s_0} \Pr(s_0) V_{\pi,\chi}(s_0)
$$

$$
= \max_\pi \min_\chi J(\pi, \chi), \tag{58}
$$

where the objective function in (58) can be expanded as follows:

$$
J(\pi, \chi)
$$

$$
= \mathbb{E}_{s_0 \sim \Pr(s_0)} \left[ V_{\pi,\chi}(s_0) \right]
$$

$$= \sum_{s_0} \Pr(s_0) V_{\pi,\chi}(s_0)$$

$$= \sum_{s_0} \Pr(s_0) \, \mathbb{E}_{a_t \sim \pi, \rho_t \sim \chi} \left[ \sum_{t=0}^{\infty} \gamma^t r_{t+1}(s_t, a_t) \mid s_0 \right]$$

$$= \sum_{s_0} \Pr(s_0) \sum_{t=0}^{\infty} \mathbb{E}_{a_t \sim \pi, \rho_t \sim \chi} \left[ \gamma^t r_{t+1}(s_t, a_t) \mid s_0 \right]$$

(linearity of the expectation)

$$= \sum_{s_0} \Pr(s_0) \sum_{t=0}^{\infty} \sum_{s_t \in \mathcal{S}} \sum_{a_t \in \mathcal{A}} \sum_{\rho_t \in \mathcal{P}} p^{\pi,\chi,s_0}(s_t) \times$$

$$\pi(a_t | \rho_t) \chi(\rho_t | s_t) \gamma^t r_{t+1}(s_t, a_t)$$

$$= \sum_{s_0} \Pr(s_0) \sum_{t=0}^{\infty} g_t^{s_0}(\pi, \chi)$$

$$= \sum_{s_0} \Pr(s_0) h^{s_0}. \tag{59}$$

Because $J(\pi, \chi)$ is a linear combination of $\{h^{s_0}\}_{s_0 \in \mathcal{S}}$, $\mathcal{S}$ is finite, and $h^{s_0}$ is continuous on $\Delta(\mathcal{A}) \times \Delta(\mathcal{P})$ for any $s_0 \in \mathcal{S}$ according to Lemma C.9, the objective function $J(\pi, \chi) = \sum_{s_0} \Pr(s_0) h^{s_0}$ is continuous on $\Delta(\mathcal{A}) \times \Delta(\mathcal{P})$. Consider the function $F(\pi) = \min_\chi J(\pi, \chi)$. Since the adversary policy space $\Delta(\mathcal{P})$ is compact, the function $F$ is continuous in $\pi$. Meanwhile, the agent policy space $\Delta(\mathcal{A})$ is closed. Therefore, there exists an agent policy $\pi$ to maximize $F$ according to the extreme value theorem.

$\square$

Theorem 4.11 shows the existence of a robust agent policy. Different from the definitions of the state-robust totally optimal agent policy and robust total Nash equilibrium, the worst-case expected state value objective does not require the optimality condition to hold for all states. Agents won't get stuck in trade-offs between different states, therefore, we can find a robust agent policy to maximize the worst-case expected state value for the SAMG problem.

Now look back at the two-agent two-state game in Fig. 8. If we use the worst-case expected state value concept from Definition 4.8 and assume that the initial state is always $s_2$, then we can conclude that $\pi_1 : \pi^1(a_1|s_1) = \pi^1(a_1|s_2) = \pi^2(a_2|s_1) = \pi^2(a_2|s_2) = 1$ is a robust agent policy, as it gives the maximum worst-case expected state value of 100 for this game.

## D Robust Multi-Agent Adversarial Actor-Critic (RMA3C) Algorithm

In general, it is challenging to develop algorithms that compute optimal or equilibrium policies for MARL under uncertainties (Zhang et al., 2020b; 2021). Our algorithm adopts centralized training and decentralized execution paradigm following the popular framework in Lowe et al. (2017). During training, there is a centralized critic $Q(s, a)$ that records the total expected return given the global state $s$ and global action $a$. The connection between $Q(s, a)$ and $V(s)$ is that for any $i \in \mathcal{N}, s \in \mathcal{S}, a \in \mathcal{A}$,

$$Q(s, a) = r(s, a) + \gamma \sum_{s' \in \mathcal{S}} p(s'|s, a) V(s'). \tag{60}$$

Each agent's state input for the actor is perturbed by an adversary $\chi^i(\cdot|s) : \mathcal{S} \to \Delta(\mathcal{P}_s^i)$. During execution, each agent $i$ selects action $a^i$ based on the perturbed state $\rho^i \in \mathcal{S}$ using a trained policy $\pi^i : \mathcal{S} \to \Delta(\mathcal{A}^i)$. We want to find a policy $\pi^i$ for each agent to maximize the worst-case expected state value in Definition 4.8 under adversarial state perturbations.

As shown in Alg. 1, our algorithm has a centralized critic network $Q$ for training. Each agent has one actor network $\pi^i$ and one adversary network $\chi^i$. The critic $Q$ takes in the true global state and global action

---

**Algorithm 1:** Robust Multi-Agent Adversarial Actor-Critic (RMA3C) Algorithm

---

**1** Randomly initialize the critic network $Q$, the actor network $\pi^i$, and the adversary network $\chi^i$ for each
    agent;
**2** Initialize target networks $Q', \pi^{i\prime}, \chi^{i\prime}$;
**3 for** *each episode* **do**
**4**     The initial state $s_0 \leftarrow$ sample from $\Pr(s_0)$;
**5**     Initialize a random process $\mathcal{X}$ for action exploration;
**6**     **for** *each time step* **do**
**7**        **for** *i=1 to n* **do**
**8**           $\rho^i \leftarrow$ sample from $\chi^i(\cdot|s)$;
**9**           $a^i \leftarrow$ sample from $\pi^i(\cdot|\rho^i) + \mathcal{X}$;
**10**        **end**
**11**        Execute actions $a = (a^1, ..., a^n)$;
**12**        Obtain the reward $r$ and the next state $s'$;
**13**        $\mathcal{D} \leftarrow \mathcal{D} \cup (s, a, r, s')$;
**14**        $s \leftarrow s'$;
**15**        $Q \leftarrow \text{MGD\_Optimizer}(Q, \mathcal{D}, Q', \pi', \chi')$;
**16**        `/* Mini-batch gradient descent optimizer for critic.`     `*/`
**17**        $\pi, \chi \leftarrow \text{GDA\_Optimizer}(Q, \pi, \chi)$;
**18**        `/* Gradient descent ascent optimizer for policies.`     `*/`
**19**        Update all target networks: $\theta^{i\prime} \leftarrow \tau\theta^i + (1 - \tau)\theta^{i\prime}$.
**20**     **end**
**21 end**

---

during the training process. It returns a $Q$-value denoting the total expected return given $s$ and $a$. The state transition experience is represented by $(s, a, r, s')$ where $s'$ is the next state. It is stored in a replay buffer $\mathcal{D}$ for the critic network's training. We apply "replay buffer" and "target network" techniques (Mnih et al., 2015). The critic network is trained with a mini-batch gradient descent optimizer in line 15. In line 16, we use Gradient Descent Ascent (GDA) optimizer (Lin et al., 2020b) to update parameters for each agent's actor network and adversary network for the maximin problem $\max_\pi \min_\chi \sum_{s_0} \Pr(s_0) V_{\pi,\chi}(s_0)$ in Theorem 4.10. A detailed introduction for the GDA optimizer is included in Appendix E.4.

We have added an adversarial network that inputs the true state and outputs a perturbed state in RMA3C. This is in contrast to MADDPG and MAPPO, which do not include such a network. Compared to M3DDPG, which has a target policy network for each agent with outputs for the action space, our adversarial network's output pertains to the state space, indicating a different computational load.

# E    Implementation Detail

All hyperparameters used in experiments are listed in table 3.

## E.1    Environments

We have tested our algorithm in environments provided by Lowe et al. (2017) as shown in Fig. 10.

### E.1.1    Cooperative navigation (CN)

This is a cooperative task. There are 3 agents and 3 landmarks. Agents want to occupy/cover all the landmarks. They need to cooperate through physical actions about their preferred landmark to cover. Also, they will be penalized when collisions happen.

Table 3: Hyperparameters for our RMA3C algorithm and the baselines.

| Parameter | RMA3C | M3DDPG | MADDPG | MAPPO |
|---|---|---|---|---|
| optimizer for the critic network | Adam | Adam | Adam | Adam |
| learning rate for agent policy $\pi$ | 0.01 | 0.01 | 0.01 | 0.0007 |
| learning rate for adversary policy $\chi$ | 0.001 | / | / | / |
| discount factor | 0.95 | 0.95 | 0.95 | 0.99 |
| replay buffer size | $10^6$ | $10^6$ | $10^6$ | / |
| activation function | Relu | Relu | Relu | Relu |
| number of hidden layers | 2 | 2 | 2 | 1 |
| number of hidden units per layer | 64 | 64 | 64 | 64 |
| number of samples per minibatch | 1024 | 1024 | 1024 | 1 |
| target network update coefficient $\tau$ | 0.01 | 0.01 | 0.01 | / |
| GDA optimizer steps | 20 | / | / | / |
| radius $d$ | 1.0 | / | / | / |
| uncertainty level $\lambda$ | 0.5 | 0.5 | 0.5 | 0.5 |
| upper boundary $u$ | 1.0 | 1.0 | 1.0 | 1.0 |
| lower boundary $l$ | -1.0 | -1.0 | -1.0 | -1.0 |
| episodes in training | 10k | 10k | 10k | 10k |
| time steps in one episode | 25 | 25 | 25 | 25 |

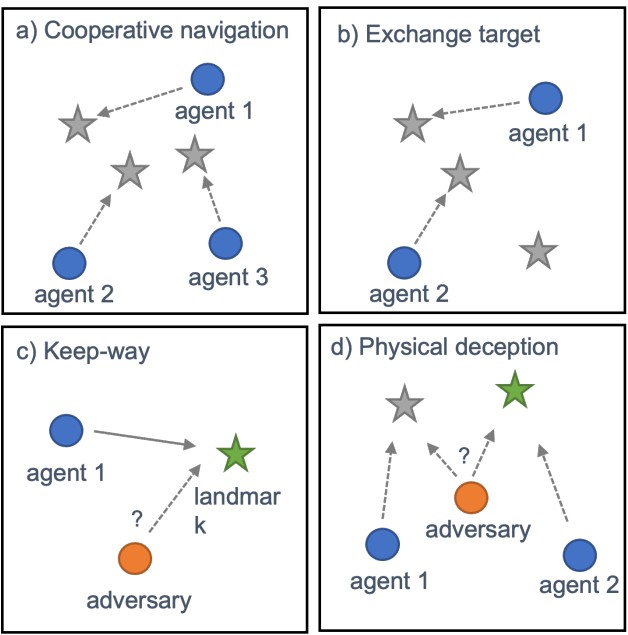

Figure 10: Some environments to test our algorithm, including a) Cooperative navigation (CN) b) Exchange target (ET) c) Keep-away (KA) d) Physical deception (PD).

### E.1.2 Exchange target (ET)

This is a cooperative task. There are 2 agents and 3 landmarks. Each agent needs to get to its target landmark, which is known only by another agent. They have to learn communication and get to landmarks. Besides, both of them are generous agents that pay more attention to helping others, i.e. rewarded more if the other agent gets closer to the target landmark.

### E.1.3 Keep-away (KA)

This is a competitive task. There is 1 agent, 1 adversary, and 1 landmark. The agent knows the position of the target landmark and wants to reach it. The adversary does not know the target landmark and wants to prevent the agent from reaching the target by pushing them away or occupying the target temporarily.

### E.1.4 Physical deception (PD)

This is a mixed cooperative and competitive task. There are 2 collaborative agents, 2 landmarks including a target, and 1 adversary. Both the collaborative agents and the adversary want to reach the target, but only collaborative agents know the correct target. The collaborative agents should learn a policy to cover all landmarks so that the adversary does not know which one is the true target.

### E.2 Baselines

We compare the performance of our algorithm with MADDPG (Lowe et al., 2017), M3DDPG (Li et al., 2019), and MAPPO (Yu et al., 2022) and follow their open-source implementation. We have a brief introduction of these methods in the following sections. There is no robustness considered in MADDPG and MAPPO. The M3DDPG considers the robustness of training partner's policies, but it does not consider state uncertainty. The MAPPO is the multi-agent version of the Proximal Policy Optimization (PPO), a popular policy gradient algorithm. Because MAPPO only works in fully cooperative tasks, we only report its results in cooperative navigation and exchange target. Note that MAPPO is also used in Guo et al. (2020) but they do not provide an open-source implementation. Therefore, we select the latest implementation in Yu et al. (2022) with the open-source code.

### E.3 Multi-Agent Deep Deterministic Policy Gradient (MADDPG)

It is difficult to apply single-agent RL algorithms directly to the multi-agent case because the environment's state transition is also influenced by the policy of other agents and it is non-stationary from a single agent's view. To alleviate this problem and stabilize training, the MADDPG algorithm is proposed using a centralized $Q$ function that has global state and global action information (Lowe et al., 2017). It assumes all agents are self-interested and every agent's objective is to maximize its own total expected return. The objective for agent $i$ is $J(\theta^i) = \mathbb{E}[R^i]$ and its gradient is

$$\nabla_{\theta^i} J(\theta^i) = \tag{61}$$
$$\mathbb{E}_{\mathbf{x},a\sim\mathcal{D}}\left[\nabla_{\theta^i}\mu^i(o^i)\nabla_{a^i}Q^i(\mathbf{x}, a^1, ..., a^n)|_{a^i=\mu^i(o^i)}\right],$$

where $Q^i(\mathbf{x}, a^1, ..., a^n)$ is a centralized action-value function, $\mathbf{x} = (o^1, \ldots, o^n)$, and $o^i$ represents agent $i$'s observation. The experience replay buffer $\mathcal{D}$ contains transition experience $\mathbf{x}, a^1, ..., a^n, \mathbf{x}', r^1, ..., r^n$ to decorrelate data. The centralized $Q^i$ can be trained using the Bellman loss:

$$\mathcal{L}(\theta^i) = \mathbb{E}_{\mathbf{x},a,\mathbf{r},\mathbf{x}'\sim\mathcal{D}}[y - Q^i(\mathbf{x}, a^1, ..., a^n)]^2,$$
$$y = r^i + \gamma Q^{i\prime}(\mathbf{x}', a^{1\prime}, ..., a^{n\prime})|_{a^{j\prime}=\mu^{j\prime}(o^j)}, \tag{62}$$

where $Q^{i\prime}$ is the target network whose parameters are copied from $Q$ with a delay to stabilize the moving target. Note that this algorithm adopts a centralized training and decentralized execution paradigm. When testing, each agent can only access its local observation to select actions.

In M3DDPG (Li et al., 2019), the uncertainty from the training partner's policies is considered: all other partners are considered as adversaries that select actions to minimize the total expected return of the training agent. In other words, when updating both actor and critic, they select training partner's actions by $a^{j\neq i} = \arg\min_{a^{j\neq i}} Q^i(\mathbf{x}, a^1, ..., a^n)$.

### E.4 Gradient Descent Ascent (GDA)

Gradient Descent Ascent (GDA) (Lin et al., 2020b) is currently one widely-used algorithm for solving the following minimax optimization problem:

$$\min_x \max_y \ f(x, y). \tag{63}$$

GDA simultaneously performs gradient descent update on the variable $x$ and gradient ascent update on the variable $y$ according to (64) with step sizes $\eta_x$ and $\eta_y$.

$$
\begin{aligned}
x_{t+1} &= x_t - \eta_x \nabla_x f(x_t, y_t), \\
y_{t+1} &= y_t + \eta_y \nabla_y f(x_t, y_t).
\end{aligned}
\tag{64}
$$

It has a variety of variants to accommodate different types of geometries of the minimax problem, such as convex-concave geometry, nonconvex-concave geometry, nonconvex-nonconcave geometry, etc.

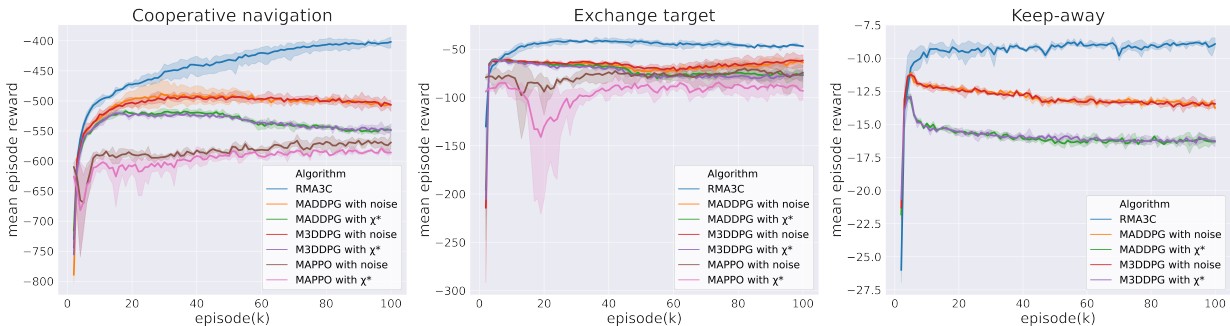

Figure 11: Our RMA3C algorithm compared with several baseline algorithms during the training process. The results showed that our RMA3C algorithm outperforms the baselines, achieving higher mean episode rewards and displaying greater robustness to state perturbations. The baselines were trained under either random state perturbations or a well-trained adversary policy $\chi^*$ (adversaries that are trained for the maximum training episodes in RMA3C). Overall, our RMA3C algorithm achieved up to 58.46% higher mean episode rewards than the baselines.

### E.5 Training Comparison Under different Perturbations

We first compare our algorithm with baselines during the training process to show that our RMA3C algorithm can outperform baselines to get higher mean episode rewards under different state perturbations. Note that our RMA3C algorithm has a built-in adversary to perturb states, so we do not train it under random state perturbations. Comparing RMA3C to other baselines with different state perturbations, the RMA3C gets higher mean episode rewards. It shows our RMA3C algorithm is more robust under different state perturbations. Comparing each baseline with random state perturbations to the same baseline with the well-trained adversary policy $\chi^*$, we can see the adversary trained by the RMA3C is more powerful than the random state perturbations. Because the adversary policy $\chi^*$ intentionally selects state perturbations to minimize agents' total expected return. The mean episode reward of the last 1000 episodes during training is shown in Table 4. Our RMA3C algorithm achieves up to 58.46% higher mean episode rewards than the baselines under different state perturbations.

### E.6 Cooperative Navigation With 6 Agents

We compare our RMA3C algorithm with baselines in the cooperative navigation scenario with more agents added. The original cooperative navigation environment has 3 agents and the training results are shown in Fig. 4. We show the training results with 6 agents in Fig. 12. After increasing the total number of agents

Table 4: Mean episode reward of the last 1000 episodes during the training. Our RMA3C algorithm achieves up to 58.46% higher mean episode rewards than the baselines. The corresponding figure is 11, and it is also included in the main content.

|  | CN | ET | KA |
|---|---|---|---|
| RMA3C (ours) | **-401.7** | **-47.02** | **-8.93** |
| MADDPG w/ $\mathcal{N}$ | -506.48 | -63.76 | -13.76 |
| M3DDPG w/ $\mathcal{N}$ | -506.54 | -61.71 | -13.45 |
| MAPPO w/ $\mathcal{N}$ | -569.07 | -94.28 | - |
| MADDPG w/ $\chi^*$ | -548.80 | -77.01 | -16.30 |
| M3DDPG w/ $\chi^*$ | -547.99 | -75.87 | -16.26 |
| MAPPO w/ $\chi^*$ | -585.83 | -113.19 | - |

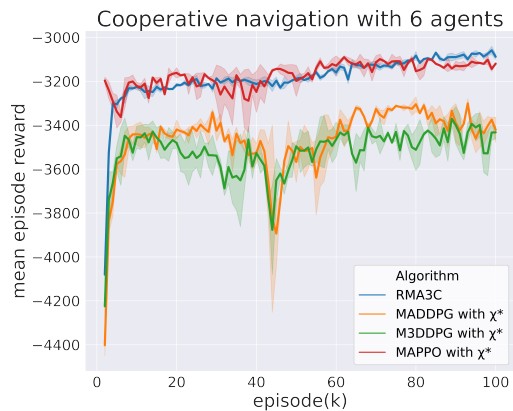

Figure 12: Our RMA3C algorithm compared with baselines during the training process in the cooperative navigation scenario with 6 agents added. Our algorithm gets higher mean episode rewards in the environment with an increased agent number.

in the environment, our RMA3C algorithm still gets higher mean episode rewards than baselines under adversarial state perturbations.

We also test the learned policies in the 6-agent Cooperative Navigation (CN) environment to show our RMA3C policy is more robust under adversarial state perturbations. During testing, the mean episode rewards are averaged across 2000 episodes and 10 test runs for each algorithm. We put all the well-trained agents using different algorithms into the 6-agent CN environment with well-trained adversary policies $\chi^*$ to perturb states. The result is shown in Table 5. Our RMA3C policy achieves up to 9.57% higher mean episode reward than the baselines with well-trained adversarial state perturbations. The result shows that our RMA3C algorithm achieves higher robustness for a multi-agent system under adversarial state perturbations.

## F  Discussions and Future Work

In this section, we add several discussions of our work as a first attempt to study different solution concepts of the SAMG problem. We also point out several future directions for the SAMG problem.

### F.1  GDA Convergence

In our RMA3C algorithm, we use Gradient Descent Ascent (GDA) optimizer (Lin et al., 2020b) to update parameters for each agent's actor network and the adversary network. Each agent updates the actor network to maximize the worst-case expected state value in Definition 4.8, while the corresponding adversary updates the adversary network to minimize the worst-case expected state value. How to solve a non-convex non-concave minimax problem is a very challenging and not yet well-solved problem. To the best of our knowledge, the GDA optimizer is currently one of the most widely used and accepted optimizers for this type of problem,

Table 5: Mean episode rewards of 2000 episodes during testing under well-trained adversarial state perturbations in the cooperative navigation environment with 6 agents. Our RMA3C policy achieves up to 9.57% higher mean episode reward than the baselines with well-trained $\chi^*$.

| Environment | CN with 6 agents |
|---|---|
| MADDPG w/$\chi^*$ | -3405.274 $\pm$ 66.18 |
| M3DDPG w/$\chi^*$ | -3452.22 $\pm$ 80.16 |
| MAPPO w/$\chi^*$ | -3121.90 $\pm$ 18.49 |
| RMA3C w/$\chi^*$ | **-3079.37 $\pm$ 16.16** |

though it is not guaranteed to always converge (Jin et al., 2020; Razaviyayn et al., 2020; Lin et al., 2020b). Our RMA3C algorithm with GDA optimizer shows performance improvement in terms of policy robustness in our experiments. Note that we only use the GDA optimizer as a tool in our algorithm by leveraging the existing literature on solving non-convex non-concave minimax problems. Future advances of numerical algorithms and solvers for this kind of minimax problem will also benefit our algorithm by replacing the GDA optimizer with new advances.

## F.2 Non-Markovian Policy

In this work, we give the first attempt to focus on the Markovian policy under adversarial state perturbations. Dealing with the non-Markovian policy will significantly complicate the problem. We are aware of the suboptimality of Markovian policies, however, considering the computational cost of the non-Markovian policy of MARL, we decide to focus on Markovian policies in this work for computational tractability. Moreover, as shown in Proposition 3.2, our SAMG problem is different from a Dec-POMDP. Considering a non-Markovian policy based on the observation-action history may not give an advantage to the agents. For example, for the two-agent two-state game in Fig. 8, if the adversary randomly perturbs the state with $\chi^i(s_1|s_2) = 0.5$ for $i = 1, 2$, then the agents still only have a 50% chance to guess the true state even with observation-action history. Considering another example for the two-agent two-state game in Fig. 8, if the adversary perturbs all states to state $s_1$ with $\chi^i(s_1|s_2) = 1$ and $\chi^i(s_1|s_1) = 1$ for $i = 1, 2$, then the agents cannot get extra information for the true state even with observation-action history. We leave the formal analysis of non-Markovian, non-stationary policy as future work.

## F.3 Non-collaborative Game

In the problem formulation, we consider a collaborative game, where all agents share one stage-wise reward function. The new objective for the SAMG, the worst-case expected state value under state perturbations, is well-defined as proved in Theorem 4.11. For non-collaborative games, if each agent has its own reward function, and adversary $i$ wants to minimize the total expected return of agent $i$, then for a fixed agent policy $\pi$, the $n$ adversaries are playing a Markov game. In this case, only the Nash equilibrium exists among $n$ adversaries, but optimal adversary policy may not exist. Therefore, for non-collaborative games, the worst-case expected state value is not well-defined. Even though the worst-case expected state value is not well-defined for non-collaborative games, the experiment results of the competitive games and mixed-cooperative-competitive game environments in Table 1 also show that our RMA3C algorithm can get larger mean episode rewards in non-collaborative games under adversarial state perturbations. Hence, our RMA3C algorithm can increase the robustness of policies of non-collaborative games in empirical experiments. We leave the formal analysis of the non-collaborative games as future work.

