# OpenReview forum: "What is the Solution for State-Adversarial Multi-Agent Reinforcement Learning?"
_TMLR — Accepted by TMLR_

### Review · Reviewer_smAS · 2023-11-09

**Summary Of Contributions:**

This paper studies the problem of state-adversarial multi-agent reinforcement learning. The authors prove that optimal agent policy and robust Nash equilibrium do not always exist in the considered setting. To address this problem, this paper introduces a new solution concept called robust agent policy and proposes a Robust Multi-Agent Adversarial Actor-Critic (RMA3C) algorithm. The experiments on multi-agent particle environments demonstrate the effectiveness of the proposed method.

**Audience:**

Yes

**Claims And Evidence:**

Yes

**Requested Changes:**

See weaknesses

Minor changes:
* Broken references on page 4 (below Prop. 3.3).
* Moving Figure 3 to the top might look more clear.

**Strengths And Weaknesses:**

I have not verified all of the proofs, but I am assuming they are correct.

Strength:
* The writing is clear and easy to follow.
* The paper is very detailed and provides a comprehensive appendix.

Weaknesses and questions:
* Current experiments are limited to the small-scale particle environments. It would further strengthen the paper if the proposed method can be evaluated in more complicated environments.
* The non-existence theorems (Theorem 4.3 and 4.7) depend on specially crafted example SAMGs. How the non-existence holds in general (e.g., with a random SAMG) is unclear.
* The code is not in the supplementary material, unlike the authors have stated in Section 5.
* In the last paragraph on page 7, why is the critic allowed to take in the true state? Shouldn't the agents be restricted to the perturbed states?
* Is the setting of global state space a bit unrealistic for multi-agent RL? Can you give some examples for this setting?

---

> ### Author Response · Authors · 2023-12-03
> **Reply to the fourth reviewer:**
>
> 1. Current experiments are limited to the small-scale particle environments. It would further strengthen the paper if the proposed method can be evaluated in more complicated environments.
>
> We acknowledge the importance of testing our approach in more diverse and complex environments. While our current experimental setup focused on particle environments due to their simplicity and controllability, MPE has been a widely applied environment in MARL literature. We plan to extend our experiments to more complex scenarios in our future work.
>
> 2. The non-existence theorems (Theorem 4.3 and 4.7) depend on specially crafted example SAMGs. How the non-existence holds in general (e.g., with a random SAMG) is unclear.
>
> As long as there is one example that shows non-existence, Theorem 4.3 and 4.7 hold. It is of independent interest. Here we give some intuition about under what circumstances optimal agent policy and robust Nash equilibrium won't exist in general: When agents have to make trade-offs between several states, the optimal agent policy or Nash equilibrium won’t exist. Because agents have to make a trade-off between several states when adversaries perturb one state to the other. We give one example when the optimal agent policy does not exist in Appendix B.5. We also provide an analysis about why Nash equilibrium does not always exist. Please check the Definition C.4 stage-wise equilibrium, Proposition C.5, and Theorem C.6 in Appendix C. In Section C.3, we explain the reason why Nash equilibrium does not always exist. This is because the agent's stage-wise equilibrium in one state may not be consistent with its stage-wise equilibrium in a different state. We also give a detailed example to illustrate this idea in Appendix C.3.
>
>
> 3. The code is not in the supplementary material, unlike the authors have stated in Section 5.
>
> We have uploaded the source code. Please download the Supplementary Material.
>
> 4. In the last paragraph on page 7, why is the critic allowed to take in the true state? Shouldn't the agents be restricted to the perturbed states?
>
> The state value function is defined based on true states and true actions as shown in Eq. (2). The connection between state value function and value function is that for any $i \in \mathcal{N}, s \in \mathcal{S}, a \in \mathcal{A}$,
> \begin{equation}
>     Q(s,a) =  r(s,a) + \gamma\sum_{s' \in \mathcal{S}} p(s'|s, a) V(s') ,
> \end{equation}
> as shown in Eq. (60). The critic (action value function) takes in the true states and actions.
>  Though adversary changes the state observation of the agents, adversary does not change the true state or state transition of the system.
>
> 5. Is the setting of global state space a bit unrealistic for multi-agent RL? Can you give some examples for this setting?
>
> It is a widely-used setting. It has been used in the following literature:
>
> Zhang, Huan, et al. "Robust deep reinforcement learning against adversarial perturbations on state observations." Advances in Neural Information Processing Systems 33 (2020): 21024-21037.
>
> Zhang, Huan, et al. "Robust reinforcement learning on state observations with learned optimal adversary." ICML (2021).
>
> Zhang, Kaiqing, et al. "Robust multi-agent reinforcement learning with model uncertainty." Advances in neural information processing systems 33 (2020): 10571-10583.
>
> He, Sihong, et al. "Robust multi-agent reinforcement learning with state uncertainty." TMLR (2023).
>
> Sun, Yanchao, et al. "Certifiably Robust Policy Learning against Adversarial Multi-Agent Communication." The Eleventh International Conference on Learning Representations. 2022.

---

> > ### Comment · Reviewer_smAS · 2023-12-16
> > **Thank you for the response**
> >
> > Sorry for the late reply and thank you for addressing my comments. I still have a few questions and would like to discuss with the authors.
> >
> > * "non-existence theorems"
> > I am not saying Thm 4.3 & 4.7 are wrong. Proving an existential statement with an example is absolutely fine and correct. My concern is that: if we don't know how ubiquitous this non-existence is, the proposed method ("another solution concept") is less well motivated.
> >
> > * "critic allowed to take in the true state"
> > If the adversary only affects the observation sent to the actor, it sounds like the critic is not viewed as part of the agent, right? This is really different from what I have known. Typically, the actor-critic agent takes in the perturbed state for both actor (policy estimation) and critic (value estimation).

---

> > > ### Author Response · Authors · 2023-12-20
> > > **Reply to Follow-up of the Fourth reviewer**
> > >
> > > We sincerely appreciate your continued engagement and insightful feedback on our manuscript. We are pleased to provide further clarification on the points raised.
> > >
> > > - Regarding "Non-existence Theorems"
> > >
> > > We recognize your concern about the example used in the proof of Theorem 4.7 in Appendix C.3. Our intention was to present a straightforward example for ease of understanding. However, we agree that constructing more counter-examples can be more illustrative in demonstrating the prevalence of non-existence scenarios. As long as the two stage-wise equilibriums have different requirements (not necessarily contrary to each other), there is no Nash equilibrium.
> > >
> > > To elaborate, consider a 2-state 2-action game. If, for state $s_1$, the stage-wise equilibrium necessitates choosing action $a^1$ with probability 0.2 and $a^2$ with probability 0.8, and for state $s_2$, the stage-wise equilibrium requires choosing $a^1$ with probability 0.6 and $a^2$ with probability 0.4, then it's clear that no Nash equilibrium can simultaneously satisfy these requirements. This example illustrates the absence of a robust Nash equilibrium in such a 2-state 2-action game scenario.
> > >
> > > Further, our experimental results underscore the necessity of an algorithm capable of handling state perturbations. As evidenced in Table 1, the mean episode rewards – averaged across 2000 episodes and 10 test runs in each environment – demonstrate significant performance drops in baseline algorithms (MADDPG, M3DDPG, and MAPPO) when random state perturbations are introduced. This performance degradation, further exacerbated in environments with well-trained adversary policies $\chi^*$ as shown in Table 2, highlights the critical need for solutions adept at managing state perturbations.
> > >
> > > Our approach, while demonstrated in specific instances of non-existence theorems, is designed to offer broader applicability. The significance of our method lies in providing alternative solutions where traditional methods may falter or be inapplicable. This versatility is crucial in advancing the field, particularly in complex scenarios where standard solutions are inadequate. We are open to suggestions on formally characterizing ubiquity and welcome further discussion on this matter. Thank you for your valuable input on this aspect.
> > >
> > > - Regarding "Critic Allowed to Take in the True State"
> > >
> > > In our model, the critic is distinct from the agent and is exclusively utilized during the training phase. This approach aligns with methodologies employed in several notable papers, including:
> > >
> > > 1. Zhang, Huan, et al. "Robust deep reinforcement learning against adversarial perturbations on state observations." Advances in Neural Information Processing Systems 33 (2020): 21024-21037.
> > > 2. He, Sihong, et al. "Robust multi-agent reinforcement learning with state uncertainty." TMLR (2023).
> > > 3. Liang, Yongyuan, et al. "Efficient adversarial training without attacking: Worst-case-aware robust reinforcement learning." Advances in Neural Information Processing Systems 35 (2022): 22547-22561.
> > >
> > > These references support our design choice and underscore its relevance in the broader context of reinforcement learning research.
> > >
> > > We hope these clarifications address your queries satisfactorily. We look forward to your feedback and are committed to further refining our work based on your guidance.

---

> > > > ### Comment · Reviewer_smAS · 2023-12-22
> > > > **Thank you**
> > > >
> > > > Thank you for clarifying. I don't have further questions. Please consider incorporating the above discussions into the next version.

---

> ### Author Response · Authors · 2023-12-22
> **Thank you! New version updated.**
>
> Thank you very much for your valuable feedback and insightful discussions. We have carefully considered your suggestions and incorporated them into the revised version of our manuscript.
>
> Specifically, we have added the following clarification after Theorem 4.11: "The robust agent policy, while motivated in specific instances of non-existence theorems, is designed to offer broader applicability. The significance of the new solution concept lies in providing alternative solutions where traditional methods may falter or be inapplicable. This versatility is crucial in advancing the field, particularly in complex scenarios where standard solutions are inadequate."
>
> Furthermore, we have expanded the explanations after the proof of Theorem 4.7 in Appendix C.3, to better articulate our reasoning:
>
> "In the proof of Theorem 4.7, we intended to present a straightforward example for ease of understanding. However, more counter-examples can be more illustrative in demonstrating the prevalence of non-existence scenarios. As long as the two stage-wise equilibriums have different requirements (not necessarily contrary to each other), there is no Nash equilibrium.
>
> To elaborate, consider a 2-state 2-action game. If, for state $s_1$, the stage-wise equilibrium necessitates choosing action $a^1$ with probability 0.2 and $a^2$ with probability 0.8, and for state $s_2$, the stage-wise equilibrium requires choosing $a^1$ with probability 0.6 and $a^2$ with probability 0.4, then it's clear that no Nash equilibrium can simultaneously satisfy these requirements. This example illustrates the absence of a robust Nash equilibrium in such a 2-state 2-action game scenario."
>
> We hope these additions enhance the clarity and depth of our manuscript. We are grateful for the opportunity to improve our work through this review process."

---

### Review · Reviewer_7VyL · 2023-11-10

**Summary Of Contributions:**

This paper studies the problem of learning a multi-agent RL policy that is robust to adversarial state perturbations. The paper shows solution concepts in this formalism must be nuanced as there is not a single policy that is optimal in every state, nor is there a set of policies that are in Nash-Equalibrium in every state. The paper proposes instead to look for a policy that has the optimal worst-case performance in expectation and proposes an algorithm to solve for it.

**Audience:**

Yes

**Broader Impact Concerns:**

There are no particularly worrying ethical implications of this work warranting a specific additional Broader Impact Statement.

**Claims And Evidence:**

Yes

**Requested Changes:**

I request that the names of the "optimal agent policy" and "Robust Nash Equilibrium" be changed to avoid confusion.

I would also recommend the Theorem statements are contextualized with references to similar proof techniques used in the study of imperfect recall in extensive form games, or other literature where relevant, so that the reader can more easily bring themselves up to speed.

**Strengths And Weaknesses:**

**Strengths:**
- The paper studies interesting nuances in using adversarial training methods in multi-agent domains
- The paper contains a collection of useful existence and non-existence results about a collection of natural solution concepts
- Overall, the paper is quite careful in its methodology
- The theorems in the paper are all correct and important for practitioners to understand.

**Weaknesses:**
- My main concern is that a few of the definitions are quite confusingly named.

Definition 4.2 is a very generic "optimal agent policy", though it is defined differently than I would expect. I would imagine the optimal agent to be one that "maximizes the worst-case value function". This makes much of the paper confusing, as the thing called "the optimal agent policy" is said not to exist, when the thing I would expect "optimal agent policy" to refer to clearly does exist.

The name used for the object in Definition 4.2 thus has to be made more specific for the paper to be easily understandable. I would suggest something like the "state-robust everywhere optimal policy" or the "state-robust totally optimal policy," which would be unambiguous.


- A similar concern holds for definition 4.6. The thing that is called the "Robust Nash Equilibrium" is not what I would call the "Robust Nash Equilibrium" since I would expect that to be "no agent can unilaterally change its policy to more effectively maximize the worse-case value function".  Again, this is confusing because the object I would call the "Robust Nash Equilibrium" does exist, and the object called the "Robust Nash Equilibrium" in the paper is shown not to. I would again suggest changing this to "Robust Everywhere Nash Equilibrium" or "Robust Total Nash Equilibrium".


- Most of the theorems in the paper would follow from viewing the SAMG as an extensive form game under imperfect recall and applying the approaches used to analyze those games. They are still useful and represent a significant amount of work, as most readers will not be familiar with those approaches and the steps are carefully walked through in the appendix. However, it would at least be useful for the reader to reference this way of thinking.

The one result that does not follow from this perspective is Theorem 4.11, which would only be easy to translate in the finite case. Essentially, by imagining the the policy as a T sampled actions for each state, and then noticing that there must exist a Nash equilibrium in this game of selecting policies -- since it is finite Nash's theorem applies. The proof technique the authors use in the infinite case I have not seen before, and I would be interested in any reference to similar cases.

**Minor Concerns:**
There are  a few typos, and missed references that make the paper more difficult to read than necessary:
"adversarial state" -> "adversarial state perturbations" in the first paragraph
- bad reference at the end of page 4, and in the proof of Theorem 4.10

---

> ### Author Response · Authors · 2023-12-03
> **Reply to the third reviewer:**
>
> 1. I request that the names of the "optimal agent policy" and "Robust Nash Equilibrium" be changed to avoid confusion.
>
>
> Thanks for the reviewer’s suggestion. We would change the names of the "optimal agent policy" and "Robust Nash Equilibrium" to be “state-robust totally optimal policy” and “Robust Total Nash Equilibrium”.
>
>
> 2. the Theorem statements are contextualized with references to similar proof techniques used in the study of imperfect recall in extensive form games, or other literature where relevant, so that the reader can more easily bring themselves up to speed.
>
>
>  We’ve added the following reference that studies imperfect recall in extensive form games. In section 4.3, we add “\blue{The theorem could also follow from viewing the SAMG as an extensive form game under imperfect recall and applying the approaches used to analyze those games~\cite{chen2012tractable}.}”
>
>
> [chen2012tractable]: Chen, Katherine, and Michael Bowling. "Tractable objectives for robust policy optimization." Advances in Neural Information Processing Systems 25 (2012).
>
> 3. We have fixed typos and the bad references at the end of page4, and in the proof of Theorem 4.10.

---

### Review · Reviewer_cWut · 2023-11-18

**Summary Of Contributions:**

To address the issue of state perturbation in multi-agent RL (MARL) frameworks, the authors propose the state-adversarial Markov games (SAMG) and investigate the state uncertainty for MARL. Specifically, they first show the non-existence of optimal agent policy and robust Nash equilibrium in SAMGs. To overcome such a difficulty, they develop the concept called robust agent policy against the state adversary, which aims to maximize the so-called worst-case expected state value. They theoretically prove the existence of robust agent policy for SAMGs and on top of this framework, designing an algorithm Robust Multi-Agent Adversarial Actor Critic (RMA3C) to learn efficiently the robust policies for all agents under state uncertainty. To validate the proposed framework and solutions, they use multi-agent particle environments and a few baselines to demonstrate the superiority and robustness.

**Audience:**

Yes

**Broader Impact Concerns:**

The authors may add one paragraph in the paper to address the broader impact concerns.

**Claims And Evidence:**

Yes

**Requested Changes:**

Please see the above weaknesses to address the concerns and issues.

**Strengths And Weaknesses:**

Strengths:

1. The investigated topic is quite interesting and relevant to the community. The paper is well written and easy to follow.
2. The paper is well motivated and looks technically sound in terms of theoretical contributions and promising empirical results.

Weaknesses:
1. Some technical detail is missing, particularly when the authors mentioned in the paper that SAMG problem cannot be solved by existing solutions for robust Markov games and existing work for single-agent RL with adversarial state perturbations. The authors may have to add some more detail about the difference between the proposed SAMG and existing works.
2. In this work, the authors assume the adversaries access the true state. Would this assumption be required always to apply SAMG? Also, in Definition 3.1, the authors include the true state inside the admissible perturbed state set. I am confused about this. Why couldn't exclude the true state from set, as long as it follows some constraints, such as the bounded distance between the true state and any perturbed state? How realistic is such a definition?
3. To prove the Proposition 4.1, the authors need to construct an MDP. Why did the authors need MDP to show a conclusion stemming originally from SAMG? This is confusing.
4. In the paper, the authors say "When agent i is calculating its robust best response, it assumes a worst-case perspective of the state perturbations." Would such an assumption be strong and why?
5. Though the experimental results look promising, the construction of the perturbed state set seems to fall back to just noise addition. But in the paper, the authors said adversary is not just noise. How did the authors construct such a set? Though we have known the difference between the true state and adversarial state is bounded by a constant $d$.

---

> ### Author Response · Authors · 2023-12-03
> **Reply to the second reviewer:**
>
> 1. The authors may have to add some more detail about the difference between the proposed SAMG and existing works.
>
> We have pointed out the difference between SAMG and existing work for single-agent RL, Dec-POMDP, and robust Markov games in Section 3. We show the connection between SAMG and Dec-POMDP, Markov game in Proposition 3.2 and Proposition 3.3. Furthermore, we give more details and examples in Appendix A “Comparison with Dec-POMDP and Markov Games”.
>
> Our SAMG problem cannot be solved by the existing work for single-agent RL with adversarial state
> perturbations (Mandlekar et al., 2017; Pattanaik et al., 2018; Zhang et al., 2020a; 2021; Liang et al., 2022).
> Each agent’s action in SAMG is selected based on its own perturbed state observation and the state knowledge
> of each agent can be different after adversarial perturbations, so the SAMG problem cannot be solved by the
> above single-agent RL where the agent has only one state observation at each stage.
>
> Our SAMG problem cannot be solved by the existing work in the Decentralized Partially Observable Markov
> Decision Process (Dec-POMDP) (Bernstein et al., 2002; Oliehoek et al., 2016) as shown in Fig. 2. In contrast,
> the policy in SAMG needs to be robust under a set of admissible perturbed states. The adversary aims to
> find the worst-case state perturbation policy χ to minimize the MARL agents’ total expected return, but the
> Dec-POMDP cannot characterize the worst-case state perturbations. Moreover, agents usually cannot get the
> true state s in Dec-POMDP, while in SAMG, the true state s is known by the adversaries. Adversaries can
> take the true state information and use it to select state perturbations for the MARL agents. The following
> proposition 3.2 shows that under a fixed adversarial policy, the SAMG problem becomes a Dec-POMDP.
> However, in SAMG the adversary policy is not a fixed policy, it may change according to the agents’ policies
> (see Theorem 4.1 for detail) and always select the worst-case state perturbation for agents. The proof of
> proposition 3.2 is in Appendix A. We also give a two-agent two-state SAMG that cannot be solved by
> Dec-POMDP in Appendix A.
>
> Proposition 3.2. When the adversary policy is a fixed policy, the SAMG problem becomes a Dec-
> POMDP (Oliehoek et al., 2016).
>
> Proposition 3.3. When the adversary policy is a fixed bijective mapping from S to S, the SAMG problem
> becomes a Markov game.
>
> Additionally, our SAMG problem cannot be solved by existing methods for robust Markov games considering
> the uncertainties from reward (Chen & Bowling, 2012; Zhang et al., 2020b), transition dynamics (Zhang et al.,
> 2020b; Hu et al., 2020; Sinha et al., 2020; Yu et al., 2021; Wang et al., 2023), training partner’s policies (Li
> et al., 2019; van der Heiden et al., 2020). These methods are not applicable to our problem because the
> agents do not have access to the true state information after adversarial perturbations.
>
> 2. the authors assume the adversaries access the true state. Would this assumption be required always to apply SAMG? Also, in Definition 3.1, the authors include the true state inside the admissible perturbed state set. I am confused about this. Why couldn't exclude the true state from set, as long as it follows some constraints, such as the bounded distance between the true state and any perturbed state? How realistic is such a definition?
>
> Yes, the adversaries always have access to the true state. We consider the set of admissible perturbed state for agent $i$ at state $s$ as $\mathcal{P}^i_s \subseteq \mathcal{S}$. We assume the admissible perturbed state set is within the state space of the agent, and the true state is also within the state space, not assuming the true state is inside the admissible perturbed state set. This is because if the perturbed state is not within the state space any more, for example, the perturbed state value is outside of the sensor’s measurement range value, then it will be easier for the detector to detect the abnormal state. It is possible to use some constraints, like the distance between the true state and any perturbed state. This is what we used in experiments. As stated in the first paragraph of experiments: “In our experiments, we consider the set of admissible perturbed state for agent $i$ at state $s$ as an $\ell_\infty$ norm ball around $s$: $\mathcal{P}^i_s = \{\rho^i \in \mathcal{S}: \| \rho^i - s \|_\infty \leq d \}$ where $d$ is a radius denoting the perturbation budget.”
>
> It is accepted as a realistic definition for studying adversaries of RL in the literature (Liu et al., 2021; Kothandaraman et al., 2021, Everett et al., 2021; Zhang et al., 2020a).

---

> ### Author Response · Authors · 2023-12-03
>
> 3. To prove the Proposition 4.1, the authors need to construct an MDP. Why did the authors need MDP to show a conclusion stemming originally from SAMG? This is confusing.
>
> The construction of an MDP in the proof of Proposition 4.1 serves as an analytical tool to prove the existence of optimal adversary policy. By constructing an MDP, it allows us to leverage existing theoretical results from MDP literature.
>
> 4. In the paper, the authors say "When agent i is calculating its robust best response, it assumes a worst-case perspective of the state perturbations." Would such an assumption be strong and why?
>
> This is a standard way to make assumptions for Nash equilibrium definitions. When an agent is finding its best response, it assumes that the other agents always take the optimal strategy against it. In the SAMG problem, the optimal strategy for the adversaries is defined as worst-case adversaries as stated in section 4.1.
>
> This way to make assumptions for best responses has been widely used in literature: The Nash equilibrium is a widely used solution concept in game theory, first proposed by Nash in Nash (1951) for general-sum finite one-shot games. It states that each player selects the best response strategy to the others’ strategies and no player would want to deviate from the equilibrium, as doing so would result in a worse utility. This concept was later extended to infinite games by Debreu (Debreu, 1952), Glicksberg (Glicksberg, 1952), and Fan (Fan, 1952). Markov games, which involve a sequential decision process in a two-player zero-sum setting, were first defined by Shapley in Shapley (1953). Fink extended the Nash equilibrium concept to Markov games in Fink (1964) and proved that an equilibrium point exists in n-player general-sum discounted Markov games. The uncertainty in transition dynamics of a Markov game was considered in Nilim & El Ghaoui (2005); Iyengar (2005) using a robust optimization approach, with independent proofs for the existence of the equilibrium point. Additionally, uncertainty in utility (or "reward" in reinforcement learning) was also taken into account in Kardeş et al. (2011) for n-player finite state/action discounted Markov games, with a proof for the existence of the equilibrium point.
>
> 5. Though the experimental results look promising, the construction of the perturbed state set seems to fall back to just noise addition. But in the paper, the authors said adversary is not just noise. How did the authors construct such a set? Though we have known the difference between the true state and adversarial state is bounded by a constant .
>
> We want to clarify that the worst-case adversaries have obvious differences from noise. In our experiments, we consider the set of admissible perturbed state for agent $i$ at state $s$ as an $\ell_\infty$ norm ball around $s$: $\mathcal{P}^i_s = \{\rho^i \in \mathcal{S}: \| \rho^i - s \|_\infty \leq d \}$ where $d$ is a radius denoting the perturbation budget.  The value of $d$ can be selected as a large value beyond the usual random noise level. The perturbation budget is used to limit the power of adversaries. Adversaries can select the best perturbation within the admissible perturbed state set to attack MARL agents. However, noise in general does not have a policy to select the best strategy to play against agents.

---

### Review · Reviewer_Gvvo · 2023-11-19

**Summary Of Contributions:**

This paper studies multi-agent systems under adversarial perturbations. The authors propose new notions of solutions, analyze them and then provide numerical experiments on several problems, including comparisons with several baselines from the literature. Overall, the main ideas are relatively well explained, although some details need to be clarified.

**Audience:**

Yes

**Broader Impact Concerns:**

No particular concerns.

**Claims And Evidence:**

Yes

**Requested Changes:**

Questions:

- (1) Page 5, equation (2): Could you please clarify how $s$ is taken into account? Does $\bar{V}_\pi$ depend on $s$? Or do you take the minimum over all states $s$?m

- (2) Section 5: For the numerical experiments, could you please comment on the computational time of each method? Also, how did you choose the hyperparameters for the baselines? Did you sweep over hyperparameters to make sure you choose the values giving the best possible results?

Typos:
- Page 4: (??)
- Page 8: “to ensure noise compact”

**Strengths And Weaknesses:**

Strengths:
- The question studied in the paper is interesting because having robust MARL methods is of importance for applications.

- There are both theoretical results and numerical experiments.

- Generally speaking, the paper is well written.

Weaknesses:

- Some details need to be clarified.

---

> ### Author Response · Authors · 2023-12-03
> **Reply to the first reviewer:**
>
> 1. How $s$ is taken into account in (2)? Does $\bar{V}_{\pi}$ depend on $s$? Or do you take the minimum over all state $s$?
>
> We thank the reviewer for pointing out this question. There are typos in (2) and (3). We have corrected them as:
>
> For a fixed agent policy $\pi$, define the worst-case state value function $\bar{V}_{\pi}$ under $\pi$ by
>
> \begin{equation}
>     \bar{V}_{\pi} (s) \defeq \min_{\chi} V_{\pi, \chi}(s)
> \end{equation}
>
> (Please check the updated manuscript for equations if it is not shown in equation mode on open review)
> for all $s \in \mathcal{S}$. An adversary policy $\chi^*$ is said to be \textit{optimal} against an agent policy $\pi$ if
> \begin{equation}
>     V_{\pi, \chi^*}(s) = \bar{V}_\pi (s)
> \end{equation}
> for all $s \in \mathcal{S}$.
>
> The worst-case value function depends on $s$. Equation (2) is defined for all $s \in \mathcal{S}$.
>
> 2. For the numerical experiments, could you please comment on the computational time of each method? Also, how did you choose the hyperparameters for the baselines? Did you sweep over hyperparameters to make sure you choose the values giving the best possible results?
>
> All methods’ computational time depends on the number of episodes and the number of steps in each episode, as shown in line 3 and line 6 in Algorithm 1 in Appendix D.
>
> We use the hyperparamers for the basedlines from their original implementation: MADDPG in Lowe et al. (2017), M3DDPG in Li et al. (2019), MAPPO in Yu et al. (2022). They all provide their source code for us to run experiments. Meanwhile, our RMA3C algorithm keep almost all the overleaping hyperparameters the same with the baselines, as shown in the hyperparameter table (Table 3) in Appendix D. We do not sweep over hyperparameters.
>
> 3. We have corrected the following typos:
>
> Page 4: (??) -> (Chen & Bowling, 2012; Zhang et al., 2020b)
>
> Page 8: “to ensure noise compact” -> “to ensure noise's compactness”

---

> > ### Comment · Reviewer_Gvvo · 2023-12-31
> > **Follow-up**
> >
> > Thank you for your reply. About point 2, I still believe that comparing algorithms solely based on the number of episodes is not a very fair comparison (or at least it does not give the full picture) and it would be important to say something about the computational time of one episode. For example, you could include the average computational time per episode for each method, to give an idea. Also, hyperparameters choices are generally problem dependent but here the values seems to be fixed arbitrarily. I think it would be good to provide some evidence that if the baselines are performing less well than your algorithm, it is not just due to a poor choice of hyperparameters.

---

> > > ### Author Response · Authors · 2024-01-06
> > > **Reply to Follow-up**
> > >
> > > Thank you for your continued engagement and valuable feedback.
> > >
> > > 1. **Computational Time and Minibatch Size**: Regarding the computational time of one step in an episode, it varies with the minibatch size. For our experiments, for a minibatch size of 1, steps 15 and 17 in our algorithm are computed once. Additionally, the computational speed is influenced by the neural network's architecture and the total number of weights.
> > > 2. **Comparative Computational Effort in RMA3C**: To address your query about the extra computational effort in the RMA3C algorithm compared to the baselines, we have added an adversarial network that inputs the true state and outputs a perturbed state. This is in contrast to MADDPG and MAPPO, which do not include such a network. Compared to M3DDPG, which has a target policy network for each agent with outputs for the action space, our adversarial network's output pertains to the state space, indicating a different computational load.
> > > 3. **Hyperparameters Selection Justification**: The hyperparameters were selected based on their implementation in foundational papers (MADDPG in Lowe et al. 2017, M3DDPG in Li et al. 2019, MAPPO in Yu et al. 2022). This choice is aligned with recent research in the field, such as Yu et al. (2022), He et al. (2023), and Bukharin et al. (2023), ensuring consistency and comparability with existing work. Adopting a uniform set of hyperparameters across studies aids in advancing the field by providing a standardized comparison framework.
> > >
> > > [Yu et al. (2022)]: Yu C, Velu A, Vinitsky E, et al. The surprising effectiveness of ppo in cooperative multi-agent games[J]. Advances in Neural Information Processing Systems, 2022, 35: 24611-24624. (From the baseline track of NeurIPS 2022)
> > >
> > > [He et al. (2023)]: He, Sihong, et al. "Robust multi-agent reinforcement learning with state uncertainty." TMLR (2023).
> > >
> > > [Bukharin et al. (2023)] Bukharin, Alexander, et al. "Robust Multi-Agent Reinforcement Learning via Adversarial Regularization: Theoretical Foundation and Stable Algorithms." Advances in Neural Information Processing Systems, (2023). （Most recent paper we saw during NeurIPS 2023 at New Orleans)
> > >
> > > 4. **Algorithm Robustness and Hyperparameter Performance**: If an algorithm performs well under normal conditions but its performance deteriorates with state perturbations, it suggests a lack of robustness to hyperparameter changes. In contrast, our RMA3C algorithm demonstrates enhanced performance under the same hyperparameters, even with state perturbations. This underscores its robustness and effectiveness compared to the baselines.
> > >
> > > We hope this clarifies the computational aspects and the rationale behind our hyperparameter choices. We look forward to further improving our manuscript based on your constructive feedback.

---

### Decision · Action_Editor_hZWo · 2023-12-27

**Recommendation:** Accept with minor revision

**Comment:**

The authors greatly improved the manuscript during the review process. All the reviewers agree, and I concur, that the is a great fit for TMLR. For the camera-ready version the authors should clarify the experimental details, specifically related to the computational cost discussion and reviewer Gvvo's comments, which did not fully make it into the current version of the paper.

**Audience:**

The paper is of interest to the MARL and control theory community.

**Claims And Evidence:**

This paper proposes solutions for MARL under adversarial perturbations. The paper contributes the theoretical analysis, connecting game theory with RL and controls, and offers baseline comparisons.